# Autonomous closed-loop mechanistic investigation of molecular electrochemistry via automation

Hongyuan Sheng [1,9] ✉, Jingwen Sun [1,9], Oliver Rodríguez[2,3,4], Benjamin B. Hoar[1], Weitong Zhang [5], Danlei Xiang[1], Tianhua Tang[6], Avijit Hazra [6], Daniel S. Min[1], Abigail G. Doyle [1], Matthew S. Sigman [6], Cyrille Costentin [7], Quanquan Gu [5], Joaquín Rodríguez-López [2,3,4] & Chong Liu [1,8] ✉

Electrochemical research often requires stringent combinations of experimental parameters that are demanding to manually locate. Recent advances in automated instrumentation and machine-learning algorithms unlock the possibility for accelerated studies of electrochemical fundamentals via high-throughput, online decision-making. Here we report an autonomous electrochemical platform that implements an adaptive, closed-loop workflow for mechanistic investigation of molecular electrochemistry. As a proof-of-concept, this platform autonomously identifies and investigates an *EC* mechanism, an interfacial electron transfer (*E* step) followed by a solution reaction (*C* step), for cobalt tetraphenylporphyrin exposed to a library of organohalide electrophiles. The generally applicable workflow accurately discerns the *EC* mechanism's presence amid negative controls and outliers, adaptively designs desired experimental conditions, and quantitatively extracts kinetic information of the *C* step spanning over 7 orders of magnitude, from which mechanistic insights into oxidative addition pathways are gained. This work opens opportunities for autonomous mechanistic discoveries in self-driving electrochemistry laboratories without manual intervention.

The past few decades have witnessed exciting advancements in molecular and organic electrochemistry[1–4]. Typical electrochemical research involves a myriad of tunable experimental parameters, ranging from universal chemistry variables to those specific to electrochemistry. The resultant complexity creates challenges for efficient discovery of electrochemical transformations and elucidation of reaction mechanisms, as it can be tedious or even impractical for experimentalists to explore the high dimensionality of the parameter space and enumerate all possible combinations manually[5–7]. As electroanalytical chemistry[8–11] has established stringent requirements of experimental conditions that must be met for defining probable mechanisms and performing electrokinetic analysis, there is a significant demand on how to identify parameter combinations suitable for mechanistic studies.

[1]Department of Chemistry and Biochemistry, University of California, Los Angeles, Los Angeles, CA 90095, USA. [2]Department of Chemistry, University of Illinois Urbana–Champaign, Urbana, IL 61801, USA. [3]Beckman Institute for Advanced Science and Technology, University of Illinois Urbana–Champaign, Urbana, IL 61801, USA. [4]Joint Center for Energy Storage Research (JCESR), Argonne National Laboratory, Lemont, IL 60439, USA. [5]Department of Computer Science, University of California, Los Angeles, Los Angeles, CA 90095, USA. [6]Department of Chemistry, University of Utah, Salt Lake City, UT 84112, USA. [7]Université Grenoble Alpes, DCM, CNRS, 38000 Grenoble, France. [8]California NanoSystems Institute, University of California, Los Angeles, Los Angeles, CA 90095, USA. [9]These authors contributed equally: Hongyuan Sheng, Jingwen Sun. ✉e-mail: hsheng7@g.ucla.edu; chongliu@chem.ucla.edu

There have been key developments in high-throughput experimentation hardware for synthetic purposes[12–17] that can be translated to electrochemical research[18–24], yet additional automated electroanalytical platforms with minimized adoption barriers are needed for fundamental mechanistic investigations. Customized multi-well parallel-plate reactors[18–22] have been reported for evaluating organic electrosynthesis, and microfabricated microfluidic devices[23,24] have been reported for automated electrokinetic measurements. Nevertheless, there remains a need for automated experimentation platforms with minimized barriers to entry that resemble standard electroanalytical setups in a typical laboratory, in order to directly utilize the vast tools available to electrochemists to obtain mechanistic information[8–11].

We envisioned that the integration of standard and automated electroanalytical tools with closed-loop decision-making for the autonomous identification of desired experimental conditions could accelerate and facilitate mechanistic investigations in fundamental electrochemistry. Automated experimentation has led to the convergence with artificial intelligence[25–28], heralding the advent of self-driving platforms that iteratively design, operate, analyze, and optimize experiments to achieve a user-defined objective[29,30]. While closed-loop screening processes based on machine-learning (ML) models or Bayesian algorithms have been emerging in the engineering of lithium-ion batteries[31,32], an autonomous closed-loop process has not been demonstrated in fundamental electrochemistry. We posit that the challenges reside in the difficulty of analyzing electrochemical data for mechanistic investigations. For example, the rich mechanistic information in cyclic voltammetry (CV), a traditional electroanalytical technique[10,11], is represented by subtle features in voltammograms that are difficult to quantify with simple figures-of-merit. Manual inspection is required to discern the evolution of voltammetric responses with different parameters such as scan rate ($v$) and/or reactant concentrations. Such manual CV analysis is not compatible with automated experimentation featuring high data throughput, hence impeding autonomous closed-loop research. It is thus critical to develop a mechanistically savvy ML model that evaluates the potentially subtle CV features and transduces them into a data format compatible with automated experimentation.

In this context, we recently reported a deep-learning (DL) model based on the residual neural network (ResNet) architecture[33] that automatically distills subtle features in voltammograms and probabilistically classifies five prototypical mechanisms in molecular electrochemistry[34,35]. One important feature of our DL model[34], different from earlier attempts[36,37], is the yielded numerical propensity distributions of probable mechanisms that addressed the aforementioned challenges in automated mechanistic analysis of electrochemical data. In this regard, our DL model unlocks the possibility of enabling a proof-of-concept autonomous closed-loop process for mechanistic investigation of molecular electrochemistry (Fig. 1a).

Building upon this achievement, here we present an autonomous electrochemical platform (Fig. 1b, c) that integrates flow chemistry for sample preparation, automated electrochemical testing, DL-based voltammogram analysis, and closed-loop decision-making based on Bayesian algorithms (Fig. 1a). We demonstrate a proof-of-concept application exploring the reactivity of cobalt tetraphenylporphyrin (CoTPP) with a library of organohalide (RX) electrophiles[38,39] wherein an $EC$ mechanism, an interfacial electron transfer ($E$ step) followed by a solution reaction ($C$ step)[9,11], was identified (Fig. 1d). The generally applicable workflow efficiently explored the parameter space and discerned the presence of an $EC$ mechanism amid negative controls and outliers. Furthermore, the workflow found suitable parameter combinations and extracted the second-order kinetic rate constants of the $C$ step ($k_0$) for different RX electrophiles spanning at least 7 orders of magnitude. The autonomous platform was capable of continuous operation for up to ~50 h by experimentally examining 2520

combinations of chemical and electrochemical parameters without manual intervention. These results demonstrated the feasibility of autonomous mechanistic discoveries in molecular and organic electrochemistry in the future.

## Results

### A proof-of-concept study using an autonomous electrochemical platform based on classical electroanalytical setups

We constructed an autonomous electrochemical platform consisting of five key modules (Supplementary Notes 1 and 2): (1) flow chemistry that enables automated electrolyte formulation and disposal (Fig. 1b, c), (2) automated electrochemical testing, including automatic $iR$ compensation during CV measurements, via a modified Hard Potato[40] Python library controlling a commercial potentiostat, (3) DL-based automated CV analysis that yields numerical propensity distributions of probable mechanisms that can be readily evaluated (Fig. 1e)[34], (4) adaptive exploration of a large parameter space using a Dragonfly[41,42] Bayesian optimization package that suggests new experimental conditions toward a user-defined objective in a closed-loop manner (Fig. 1f), and (5) a conventional single-compartment electrochemical cell with a three-electrode configuration (Fig. 1b, c), ubiquitous in electrochemistry textbooks and laboratories. The use of classical electroanalytical setups ensures the consistency of future fundamental research and the validity of deploying the rich existing knowledge and literature in electrochemistry. The entire platform can be conveniently installed in a glovebox (Fig. 1b, c) to ensure compatibility with oxygen- and moisture-sensitive chemistry.

As an initial case study, we deployed an autonomous closed-loop workflow to investigate the oxidative addition of RX electrophiles to electrogenerated low-valent metal complexes, a key activation step in numerous metal-catalyzed transformations[3,4]. In a prototypical $EC$ mechanism (Fig. 1d), a quasi-reversible if not completely reversible reduction of $Co^{II}TPP$ ($E$ step) yields a nucleophilic $Co^ITPP$ species that attacks the R–X bond and forms a metal–alkyl bond ($C$ step) via a number of possible pathways[4,43]. As the nature of the electrophiles can differ greatly, an $EC$ mechanism may not be uniformly operational. Therefore, we designed a generally applicable closed-loop workflow to probe two questions autonomously (Fig. 1f): (1) without any a priori knowledge of a given electrophile's reactivity, can the workflow explore the parameter space and discern the possible presence of an $EC$ mechanism? (2) if an $EC$ mechanism is present, can the workflow suggest and identify the desired experimental conditions for measuring the rate ($k_0$) of the $C$ step? Quantification of kinetic rate constants is the foundation of mechanistic studies, and it is non-trivial to locate not one but multiple suitable combinations of both $v$ and RX concentration ([RX]) so that $k_0$ can be appropriately determined[9,44–46]. Here such mechanistic studies are shown to be autonomously completed without manual intervention.

### The scarcity of suitable parameter combinations for electrokinetic analysis, showcased by automated exhaustive experiments

We first evaluated the practical parameter space suitable for measuring $k_0$ in a model system when RX = 1-bromobutane ($n$-BuBr)[38]. When $[Co^{II}TPP] = 1$ mM in dimethylformamide (DMF) solvent using tetrabutylammonium hexafluorophosphate (0.1 M $NBu_4PF_6$) as the supporting electrolyte, the voltammograms evolved with both $v$ and [RX] (Fig. 2a). A single-electron quasi-reversible if not completely reversible $Co^{II/I}$ redox event at −1.275 V versus ferrocene/ferrocenium (Fc/Fc⁺) was observed with high $v$ and low [RX], yet the redox's irreversibility grew more prominent as $v$ decreased and [RX] increased, in accordance with the kinetic zone diagram of an $EC$ mechanism when the $C$ step can be considered irreversible with a large equilibrium constant[9,11].

Our platform, without implementing closed-loop decision-making, exhaustively sampled the parameter space and yielded

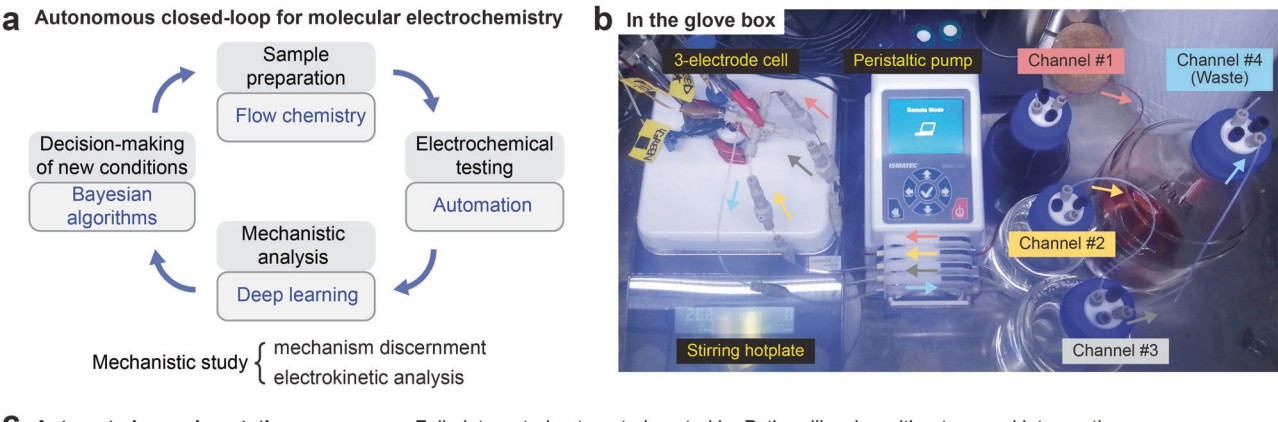

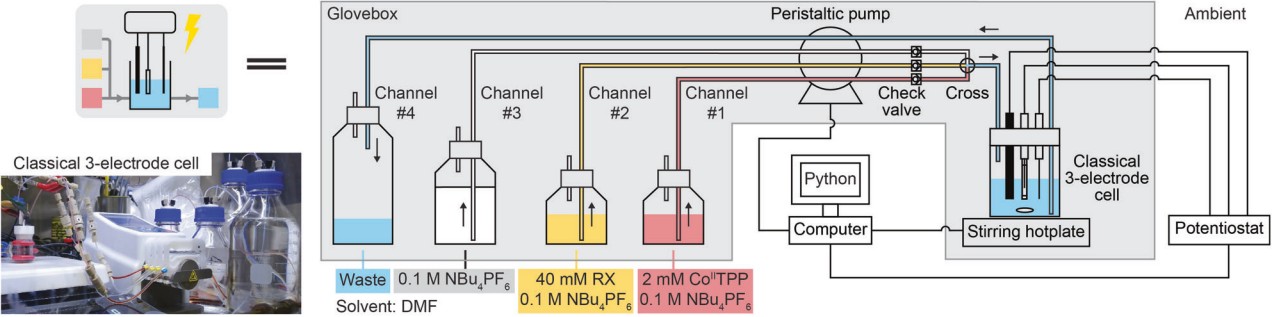

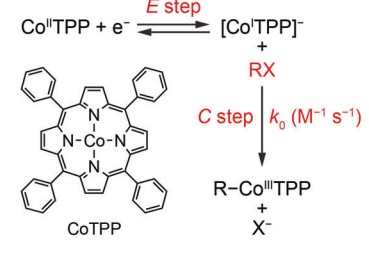

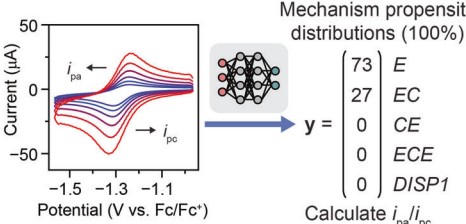

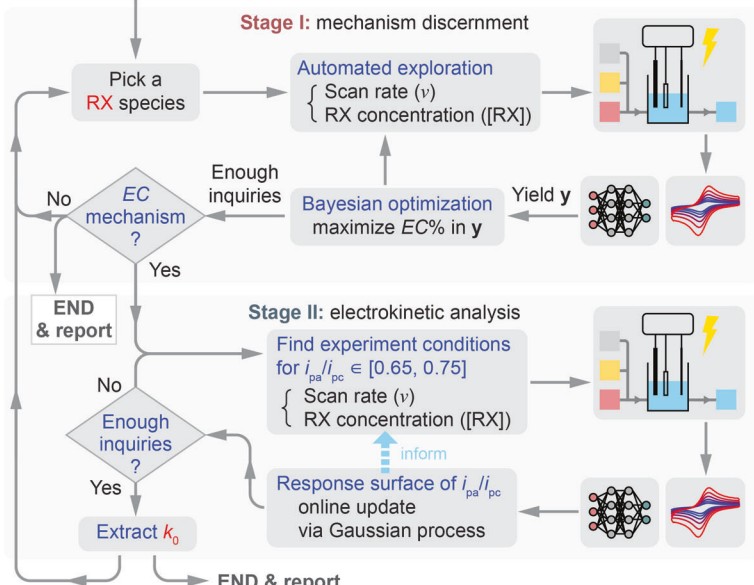

**Fig. 1 | Proof-of-concept autonomous research in fundamental electro-chemistry. a** Concept of an autonomous closed-loop process for mechanistic investigation of molecular electrochemistry from cyclic voltammetry (CV). **b** Annotated photograph and (**c**) schematic of our constructed autonomous electrochemical platform based on classical electroanalytical setups and installed in an oxygen/moisture-free glovebox. The platform is deployed for mechanistic studies of the reaction of cobalt tetraphenylporphyrin (CoTPP) with a library of organohalide (RX) electrophiles in dimethylformamide (DMF) solvent using tetrabutylammonium hexafluorophosphate (NBu$_4$PF$_6$) as the supporting electrolyte. **d** A prototypical *EC* mechanism in which a quasi-reversible if not completely reversible reduction of Co$^{II}$TPP (*E* step) yields a nucleophilic Co$^{I}$TPP species that attacks the R−X bond and forms a metal−alkyl bond (*C* step) bearing a characteristic second-order kinetic rate constant ($k_0$). **e** Deep-learning (DL) capability that transduces CV

features into quantifiable figures-of-merit (mechanism propensity distributions, denoted as **y**) compatible with downstream Bayesian optimization and automated experimentation. The voltammograms in (**e**) are adapted from the middle panel in Fig. 2a, where the experimental conditions are detailed. The reverse-to-forward peak current ratio ($i_{pa}/i_{pc}$) between anodic ($i_{pa}$) and cathodic ($i_{pc}$) peak currents of the Co$^{II/I}$ redox is defined in Supplementary Note 5. Five prototypical mechanisms (*E, EC, CE, ECE, DISP1*) in molecular electrochemistry are defined in Supplementary Table 6. **f** A generally applicable closed-loop workflow designed to explore the parameter space and discern the possible presence of an *EC* mechanism given a specific RX (Stage I) and, if an *EC* mechanism is present, to further suggest and identify the desired experimental conditions for the quantification of $k_0$ value (Stage II).

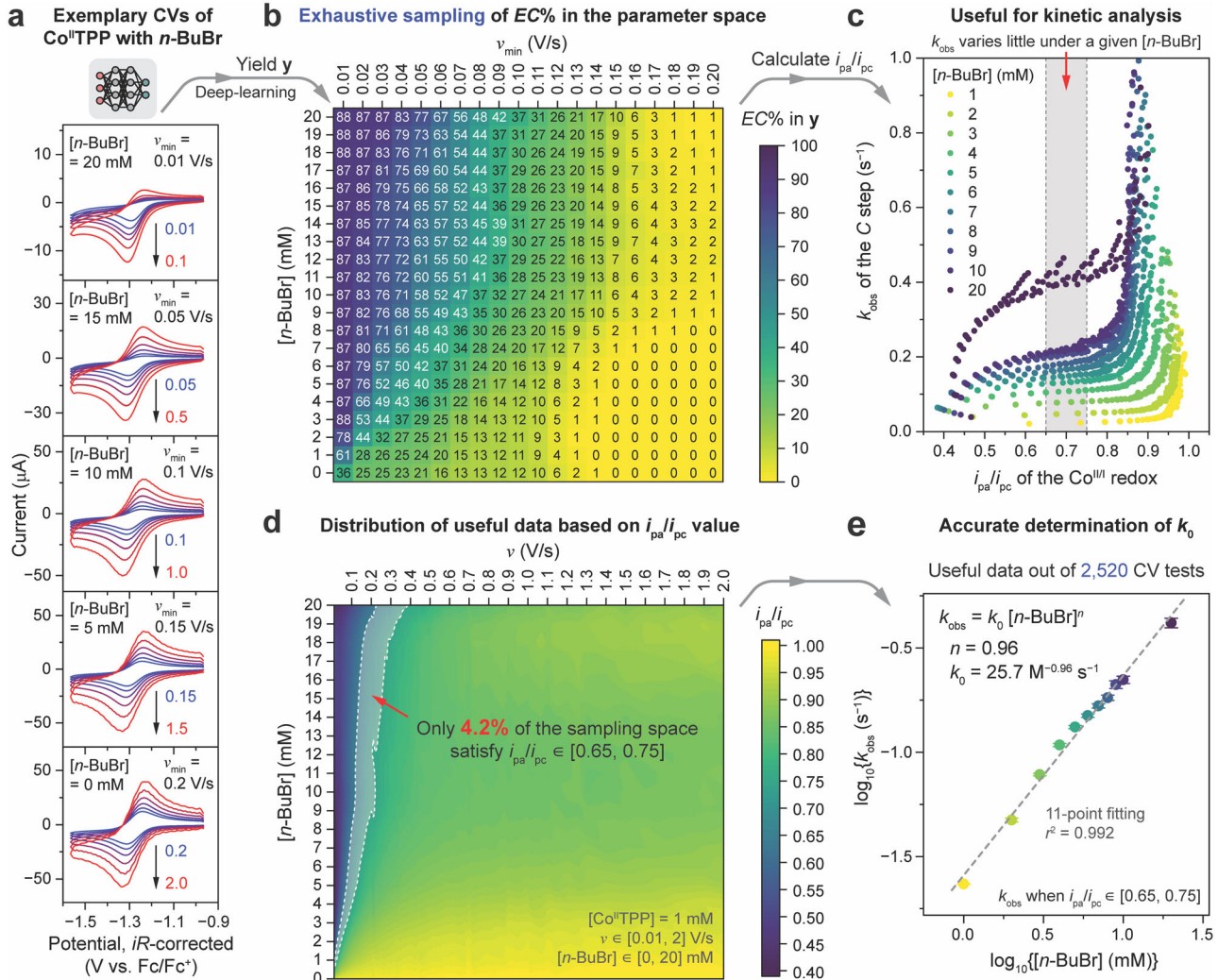

**Fig. 2 | Automated exhaustive CV experiments that illustrate the scarcity of useful data for electrokinetic analysis. a** Representative sets of six voltammograms, tested with automatic $iR$ compensation. $[Co^{II}TPP] = 1$ mM in DMF solvent using 0.1 M $NBu_4PF_6$ supporting electrolyte; RX = 1-bromobutane ($n$-BuBr) and [RX] $\in$ [0, 20] mM; six logarithmically-sampled scan rates ($v$) with the maximal and minimal $v$ differing by a factor of 10 ($v_{min}/v_{max} = 1/10$) and $v_{min} \in$ [0.01, 0.2] V/s. **b** The DL-generated propensity of an $EC$ mechanism (denoted as $EC\%$) obtained from 420 combinations, representing 2520 voltammograms, of [$n$-BuBr] and $v_{min}$ ([$n$-BuBr] $\in$ [0, 20] mM and $v_{min} \in$ [0.01, 0.2] V/s). **c** The observed pseudo-first-order kinetic rate constant of the $C$ step ($k_{obs}$) derived from and plotted versus $i_{pa}/i_{pc}$ (defined in Supplementary Note 5) of the $Co^{II/I}$ redox in the selected voltammograms with non-zero [$n$-BuBr] values, with the colors of the data points scaling with increases in [$n$-BuBr]. The shaded area in (**c**) suggests that such kinetic analysis should only be performed when $i_{pa}/i_{pc} \in$ [0.65, 0.75], as the derived $k_{obs}$ values show little variation under a given [$n$-BuBr] value and appear to grow proportionally with increases in [$n$-BuBr]. **d** The mapping of $i_{pa}/i_{pc}$ of the $Co^{II/I}$ redox in each of the 2520 voltammograms as a function of $v$ and [$n$-BuBr] ($v \in$ [0.01, 2] V/s and [$n$-BuBr] $\in$ [0, 20] mM). The shaded area in (**d**) highlights that the experimental conditions for $i_{pa}/i_{pc} \in$ [0.65, 0.75] correspond to only 4.2% of the sampling space. **e** A linear regression of $\log_{10}(k_{obs})$ versus $\log_{10}[n$-BuBr] among those valid $k_{obs}$ values derived from $i_{pa}/i_{pc} \in$ [0.65, 0.75] in (**c**), yielding the reaction order ($n$) and $k_0$ for $n$-BuBr. The colors of the data points scale with increases in [$n$-BuBr], and the vertical error bars represent the standard deviations of $\log_{10}(k_{obs})$ among all the valid $k_{obs}$ values at each [$n$-BuBr] value. Data distributions underlying the error bars in (**e**) are presented in Source Data file.

quantifiable figures-of-merit from voltammograms. Owing to the nature of our DL model[34], each inquiry in the parameter space includes a set of six voltammograms (Fig. 2a) under a given [RX] and six logarithmically-sampled $v$ values with the maximal and minimal $v$ differing by a factor of 10 ($v_{max}/v_{min} = 10$), and thus is represented as a function of both [RX] and $v_{min}$ (Fig. 2b). Setting $[Co^{II}TPP] = 1$ mM and RX = $n$-BuBr, 420 combinations of [RX] and $v_{min}$ ([RX] $\in$ [0, 20] mM and $v_{min} \in$ [0.01, 0.2] V/s, Supplementary Note 3), representing 2520 voltammograms, were automatedly prepared by flow chemistry and tested with automatic $iR$ compensation in the glovebox. The experiments took ~50 h continuously without manual intervention, at a rate of ~1.2 min per CV measurement. The 420 voltammogram sets were individually analyzed by our DL model[34] immediately after each voltammogram set was measured, yielding the propensity values of five

mechanisms (Fig. 1e; $E, EC, CE, ECE, DISP1$, defined in Supplementary Table 6). The DL analysis of such a model system[38] yielded non-zero propensity values for only $E$ and/or $EC$ mechanisms over the entire parameter space, whereas zero propensity values for $CE, ECE$, and $DISP1$ mechanisms were obtained (Supplementary Fig. 9). The DL-generated propensity of an $EC$ mechanism, extracted from voltammogram features, is higher under high [RX] and low $v_{min}$ (Fig. 2b), consistent with what a human researcher would conclude yet now a quantifiable value in lieu of a descriptive notion is provided with sufficient detection sensitivity. The obtained propensity of an $EC$ mechanism not only will discern its possible presence but also will serve as a numerical figure-of-merit for Bayesian optimization (Supplementary Note 4) to enable closed-loop decision-making (Fig. 1f).

The combinations of $v$ and [RX] values that are suitable for measuring $k_0$ are scarce. In general, $k_0$ can be derived from the observed pseudo-first-order kinetic rate constant of the $C$ step ($k_{obs}$) defined as $k_{obs} = k_0$ [RX]$^n$, where $n$ is the reaction order of RX. For any combination of $v$ and [RX] that leads to a partially irreversible Co$^{II/I}$ redox feature in a voltammogram, $k_{obs}$ can be derived from the reverse-to-forward peak current ratio ($i_{pa}/i_{pc}$, defined in Supplementary Note 5) between anodic ($i_{pa}$) and cathodic ($i_{pc}$) peak currents of the Co$^{II/I}$ redox (depicted in Fig. 1e)[44-46]. However, such kinetic analysis should only be performed when $i_{pa}/i_{pc} \in$ [0.65, 0.75] (Fig. 2c and Supplementary Note 5)[44-46], a narrow range in which the Co$^{II/I}$ redox's irreversibility is neither too strong nor too weak to ensure the validity of $k_{obs} = k_0$ [RX]$^n$. When RX = $n$-BuBr, the mapping of $i_{pa}/i_{pc}$ as a function of $v$ and [RX] (Fig. 2d) highlights that the experimental conditions for $i_{pa}/i_{pc} \in$ [0.65, 0.75] correspond to only 4.2% of the sampling space, resulting in only a modest portion of useful voltammograms out of the 2520 measured ones.

The extracted values of $k_0$ and $n$ are consistent with literature values when RX = $n$-BuBr. Figure 2c plots the $k_{obs}$ values derived from the selected voltammograms with non-zero [$n$-BuBr] values, regardless of the validity of such analysis, versus $i_{pa}/i_{pc}$. Only those $k_{obs}$ values derived from $i_{pa}/i_{pc} \in$ [0.65, 0.75] are valid, as they show little variation under a given [$n$-BuBr] value and appear to grow proportionally with increases in [$n$-BuBr] (the shaded area in Fig. 2c). Among the valid $k_{obs}$ values, an 11-point linear regression of log$_{10}$[$k_{obs}$] versus log$_{10}$[$n$-BuBr] (Fig. 2e) establishes that $k_{obs} = 25.7$ M$^{-0.96}$ s$^{-1}$ [$n$-BuBr]$^{0.96}$. The extracted $k_0$ value assuming pseudo-first-order kinetics for $n$-BuBr agrees well with a reported $k_0$ of 30 M$^{-1}$ s$^{-1}$ (ref. 38). As manual efforts of locating the desired 4.2% of the sampling space can be tedious, the scarcity of suitable parameter combinations calls for an adaptive, closed-loop search to accelerate kinetic analysis.

## Adaptive, closed-loop workflow of mechanism discernment and electrokinetic analysis

We constructed an adaptive, closed-loop workflow that involves two stages (Fig. 1f, detailed in Supplementary Fig. 16): Stage I for mechanism discernment and Stage II for electrokinetic analysis. As RX's reactivity may vary greatly, the accessible range of [RX] is expanded from [0, 20] mM to [0.008, 1000] mM with additional RX reservoirs in the flow chemistry module (Supplementary Note 6). In Stage I, Bayesian optimization[41,42] is deployed with a designated goal of maximizing the voltammograms' propensity, obtained from the DL model, toward an $EC$ mechanism by varying [RX] and $v_{min}$ (Supplementary Note 4). After a finite number of optimization steps, Stage II will be activated if the DL-generated propensity of an $EC$ mechanism can exceed 50%, otherwise the workflow terminates. In Stage II, the workflow first determines a cutoff RX concentration, [RX]$_{cutoff}$, below which avoids assessing unnecessarily high [RX] values in future inquiries (Supplementary Note 7). The workflow also creates an initial response surface of $i_{pa}/i_{pc}$ as a function of [RX] and $v$, from the existing CV data in Stage I, via a Gaussian process[41,42]. Based on the determined [RX]$_{cutoff}$ and the response surface of $i_{pa}/i_{pc}$, the workflow iteratively designs combinations of [RX] and $v$ for the next inquiry that may satisfy $i_{pa}/i_{pc} \in$ [0.65, 0.75], and automatically executes CV measurements in the glovebox. Each inquiry in Stage II includes a set of six voltammograms under an assigned [RX] value below [RX]$_{cutoff}$ and six proximate $v$ values (Supplementary Note 8). The workflow iteratively adds the resultant CV data to the cumulative database, determines whether the new data satisfy $i_{pa}/i_{pc} \in$ [0.65, 0.75], updates the response surface of $i_{pa}/i_{pc}$ on-the-fly, and initiates the next iteration of autonomous inquiry if required.

Testing such an autonomous workflow without any a priori knowledge of a given RX's reactivity validated our design. A negative control, in which acetonitrile (CH$_3$CN) was found to be unreactive toward Co$^I$TPP (Supplementary Fig. 23), confirmed that the workflow

would not indistinguishably assign an $EC$ mechanism to any substrate and would properly terminate at the end of Stage I. When RX = $n$-BuBr, a 15-step campaign of Bayesian optimization in Stage I (Fig. 3a), including 6 random sampling and 9 optimization steps, satisfactorily yielded the desired maximized propensity of an $EC$ mechanism for this model system[38] (Fig. 3b and Supplementary Fig. 24). As the maximized propensity of an $EC$ mechanism far exceeded the 50% threshold, Stage II was triggered with a determined [RX]$_{cutoff}$ = 24.3 mM (Supplementary Note 7). A response surface of $i_{pa}/i_{pc}$ was initiated (Fig. 3c), and a customized algorithm was executed to design and assess the next combinations of [RX] and $v$ for 19 iterative inquiries (Fig. 3d and Supplementary Note 8), meanwhile continuously updating the response surface of $i_{pa}/i_{pc}$ (Supplementary Fig. 25). Eventually, desired combinations of [$n$-BuBr] and $v$ that satisfy $i_{pa}/i_{pc} \in$ [0.65, 0.75] were found at 2 and 9 [$n$-BuBr] values in Stage I and II, respectively (Supplementary Fig. 26), after a total of 34 inquiries for both stages where 204 voltammograms were measured within 12 h. Subsequent linear regression of log$_{10}$[$k_{obs}$] versus log$_{10}$[$n$-BuBr] (Fig. 3e) led to $k_{obs} = 32.6$ M$^{-1.01}$ s$^{-1}$ [$n$-BuBr]$^{1.01}$, in good agreement with that determined from automated exhaustive CV experiments in Fig. 2e. The measured $k_0$ value is robust with a relative uncertainty of less than 5%, as determined by triplicating the workflow ($k_0 = 31 \pm 1$ M$^{-1}$ s$^{-1}$, Supplementary Fig. 28 and Fig. 4a). It is noteworthy that updating the response surface of $i_{pa}/i_{pc}$ on-the-fly is beneficial for successfully locating desired combinations of [RX] and $v$. The final response surface of $i_{pa}/i_{pc}$ (Fig. 3f) displayed significant differences from the initial response surface (Fig. 3c). An in silico analysis, where the response surface of $i_{pa}/i_{pc}$ was not updated, suggested a near 50% reduction in the success rate of locating the desired parameter combinations (Supplementary Note 9). Instead of exhaustively testing 2520 voltammograms for ~50 h (Fig. 2b), the autonomous closed-loop workflow reduced the number and duration of experiments by roughly a factor of 10 and 5, respectively, meanwhile assessing a 50-times wider range of [RX].

## Autonomous studies of a diverse scope of organohalide substrates for mechanistic insights

The generality of the as-described experimentation platform and closed-loop workflow (Fig. 1f and Supplementary Fig. 16) enabled autonomous investigation of an $EC$ mechanism between CoTPP and a diverse scope of RX substrates. Figure 4a summarizes the measured $k_0$ values of up to 30 substrates, over multiple orders of magnitude, toward the electrogenerated Co$^I$TPP. For one RX substrate, the autonomous workflow of 14 ± 3 h on average was executed with 33 ± 1 inquiries for both stages (or 15 inquiries if autonomously terminated after Stage I), which would otherwise take an estimated ~50 h with 420 inquiries if employing an automated exhaustive screening. If we assume 1 h for a typical human researcher to manually conduct one inquiry including sample preparation and six CV measurements, we estimate that the autonomous workflow could shorten the total time of experiments by ~60%.

The reactivity of unactivated RX substrates are, as expected, sensitive to the halogen leaving group, as $k_0$ decreases dramatically from 1-iodobutane ($n$-BuI) to $n$-BuBr to 1-chlorobutane ($n$-BuCl) whose reactivity is undetectable within the inquired parameter space (Supplementary Figs. 28-30). Meanwhile, the linear alkyl chain length has little effect on $k_0$ among $n$-BuBr, 1-bromohexane ($n$-HexBr), and 1-bromooctane ($n$-OctBr), whereas steric effects are observed with a decrease in $k_0$ when comparing $n$-BuBr to 1-bromo-2-methylpropane ($i$-BuBr) to 2-bromobutane (2-BuBr) and from $n$-BuI to neopentyl iodide (Me$_3$CCH$_2$I) (Supplementary Figs. 31-35). Also, as expected, electron-withdrawing functional groups can activate RX substrates for nucleophilic addition (Supplementary Figs. 36 and 37), as $k_0$ increases from $n$-BuCl to dichloromethane (CH$_2$Cl$_2$) to chloroacetonitrile (ClCH$_2$CN) whose measured $k_0 = 1.6 \times 10^4$ M$^{-1}$ s$^{-1}$ is consistent with a literature value ($1.9 \times 10^4$ M$^{-1}$ s$^{-1}$)[39]. Such inductive effect diminishes as the

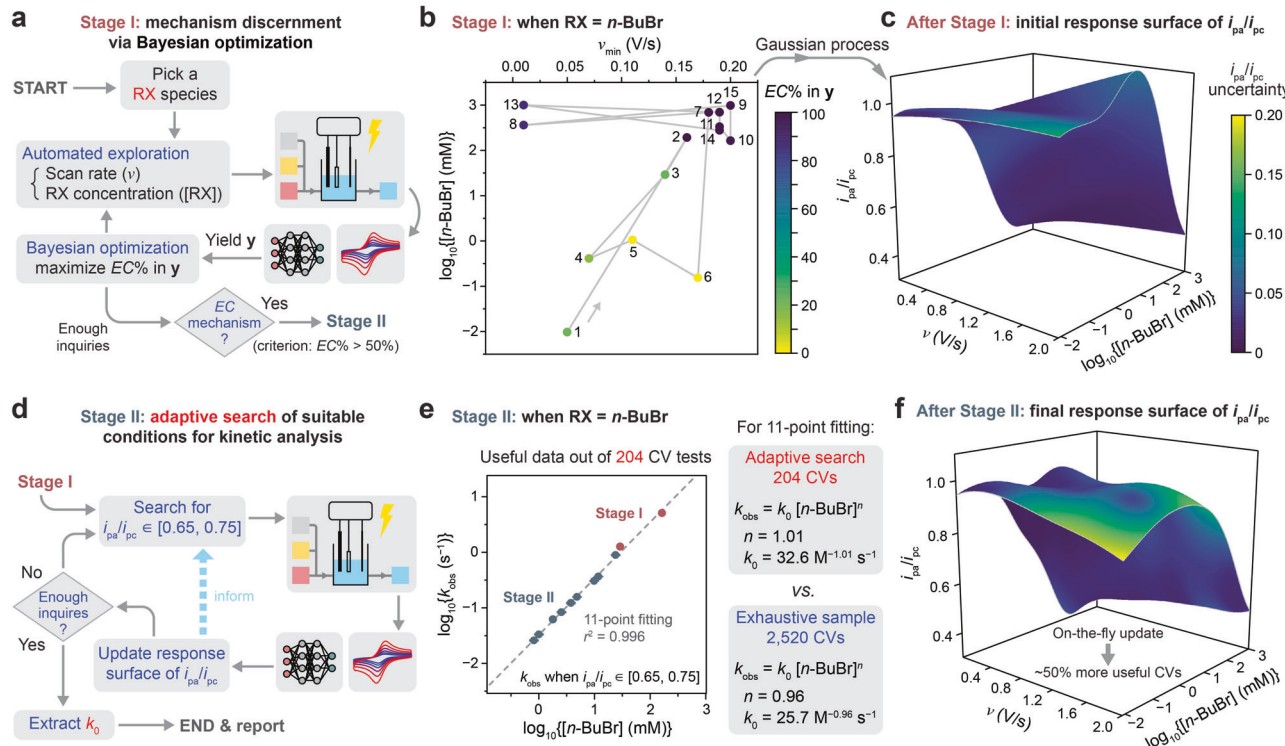

**Fig. 3 | Autonomous closed-loop workflow of mechanism discernment and electrokinetic analysis. a** The workflow of Stage I for DL-based discernment of an *EC* mechanism via Bayesian optimization. A system is considered to bear an *EC* mechanism when the maximized DL-generated propensity of an *EC* mechanism exceeds 50%. **b** The trajectory of a 15-step campaign of Bayesian optimization when RX = *n*-BuBr. [RX] ∈ [0.008, 1000] mM and $v_{min}$ ∈ [0.01, 0.2] V/s. **c** The initial response surface of $i_{pa}/i_{pc}$ as a function of $v$ and [*n*-BuBr] ($v$ ∈ [0.01, 2] V/s and [*n*-BuBr] ∈ [0.008, 1000] mM), created via a Gaussian process, after completion of Stage I. **d** The workflow of Stage II for adaptive search of parameter combinations suitable for electrokinetic analysis. **e** Desired combinations of [*n*-BuBr] and $v$ that

satisfy $i_{pa}/i_{pc}$ ∈ [0.65, 0.75] are found at 2 and 9 [*n*-BuBr] values in Stage I and II, respectively, resulting in a linear regression of $\log_{10}(k_{obs})$ versus $\log_{10}[n\text{-BuBr}]$ for the quantification of $k_0$ value. The vertical error bars represent the standard deviations of $\log_{10}(k_{obs})$ among all the valid $k_{obs}$ values at each [*n*-BuBr] value. **f** The on-the-fly updated response surface of $i_{pa}/i_{pc}$ after completion of Stage II. The colors of the surfaces in (**c**) and (**f**) represent the local model uncertainty from the Gaussian process (the standard deviation of the modeled $i_{pa}/i_{pc}$). The fluctuations in the surfaces in (**c**) and (**f**) could be ascribed to the fact that a small number of data points are fitted to a large parameter space. Data distributions underlying the error bars in (**d**) are presented in Source Data file.

electron-withdrawing group is more remote from the electrophilic carbon, which is supported by comparing the rate of ClCH$_2$CN to 3-chloropropionitrile (Cl(CH$_2$)$_2$CN) and 5-chlorovaleronitrile (Cl(CH$_2$)$_4$CN) (Supplementary Figs. 38 and 39). Lastly, the platform's low relative uncertainty for $k_0$ measurements allowed us to determine the secondary α-deuterium kinetic isotope effect (2° KIE *per* α-D): 1.25 ± 0.03 for *n*-BuBr relative to *n*-BuBr-d$_9$ (CD$_3$(CD$_2$)$_3$Br), 1.28 for *n*-BuI relative to *n*-BuI-d$_9$ (CD$_3$(CD$_2$)$_3$I) (Supplementary Figs. 40 and 41), and 1.06 ± 0.02 for primary benzyl bromide (PhCH$_2$Br) relative to PhCD$_2$Br (Supplementary Note 10 and Supplementary Figs. 45–48).

The autonomous platform was further deployed in a Hammett study of *para*-substituted primary benzyl bromide derivatives (*p*-X-PhCH$_2$Br, where X represents a *para*-substituent) (Fig. 4a and Supplementary Figs. 49–60), after a revision to the workflow to accommodate their faster rates with Co$^I$TPP (Supplementary Note 10). Such reactions are relevant to the recent interest in organic electrosynthesis catalyzed by low-valent first-row transition metals[43,47,48]. We performed linear free energy relationship analysis (Supplementary Figs. 61 and 62) using various polar- (such as $\sigma$) or radical-derived Hammett parameters[43,49] that are defined in Supplementary Table 13. In Fig. 4b, the measured $k_0$ values of ten *p*-X-PhCH$_2$Br substrates relative to that of unsubstituted PhCH$_2$Br ($\log_{10}(k_X/k_H)$) were plotted against $\sigma$. A linear correlation with a positive slope among all substrates ($r^2$ = 0.81, $\rho$ = 1.35) indicates a negative charge buildup in the transition state, consistent with the anionic nature of the Co$^I$TPP nucleophile. The observed deviation from a perfect linear correlation could be ascribed to structural variations in the transition state that have been often

observed for S$_N$2 reactions at benzylic sites[50–54], or different *para*-substituents acting to stabilize the transition state with a differing balance of inductive and resonance effects[49]. Also, the less reactive secondary benzyl bromide (PhCH(CH$_3$)Br) (Supplementary Fig. 63) compared to PhCH$_2$Br is aligned with typical S$_N$2 reactions. Empowered by the autonomous platform that accelerates experimentation, our studies of benzyl bromide substrates and beyond support a S$_N$2-type pathway (Fig. 4b), out of other possible alternative pathways[43,47,48], for the *C* step between Co$^I$TPP and RX electrophiles (Supplementary Note 11 and Supplementary Tables 14 and 15).

Intriguingly, the closed-loop workflow found detectable reactivity of 4-chlorobutyronitrile (Cl(CH$_2$)$_3$CN) toward Co$^I$TPP but autonomously terminated after Stage I, because Cl(CH$_2$)$_3$CN was deemed to not follow an *EC* mechanism by the DL model (Supplementary Fig. 65). This autonomous execution by the platform was surprising, since other similar substrates including Cl(CH$_2$)$_2$CN, Cl(CH$_2$)$_4$CN, and the more reactive 4-bromobutyronitrile (Br(CH$_2$)$_3$CN) and 4-iodobutyronitrile (I(CH$_2$)$_3$CN) (Supplementary Figs. 66 and 67) all obey an *EC* mechanism. Through additional manual CV measurements (Fig. 4c), we confirmed Cl(CH$_2$)$_3$CN as a mechanistic outlier, with the exclusion of potential artifacts including chemical purity, automated electrolyte formulation, or electrode fouling (Supplementary Note 12). An irreversible reductive shoulder peak anodic of the Co$^{II/I}$ redox appeared and gained prominence with higher [RX], accompanied with an intensified reductive peak of Co$^{II}$TPP. We hypothesize that an outer-sphere electron transfer pathway[47] involving an alkyl radical intermediate and transient regeneration of Co$^{II}$TPP (Fig. 4c and

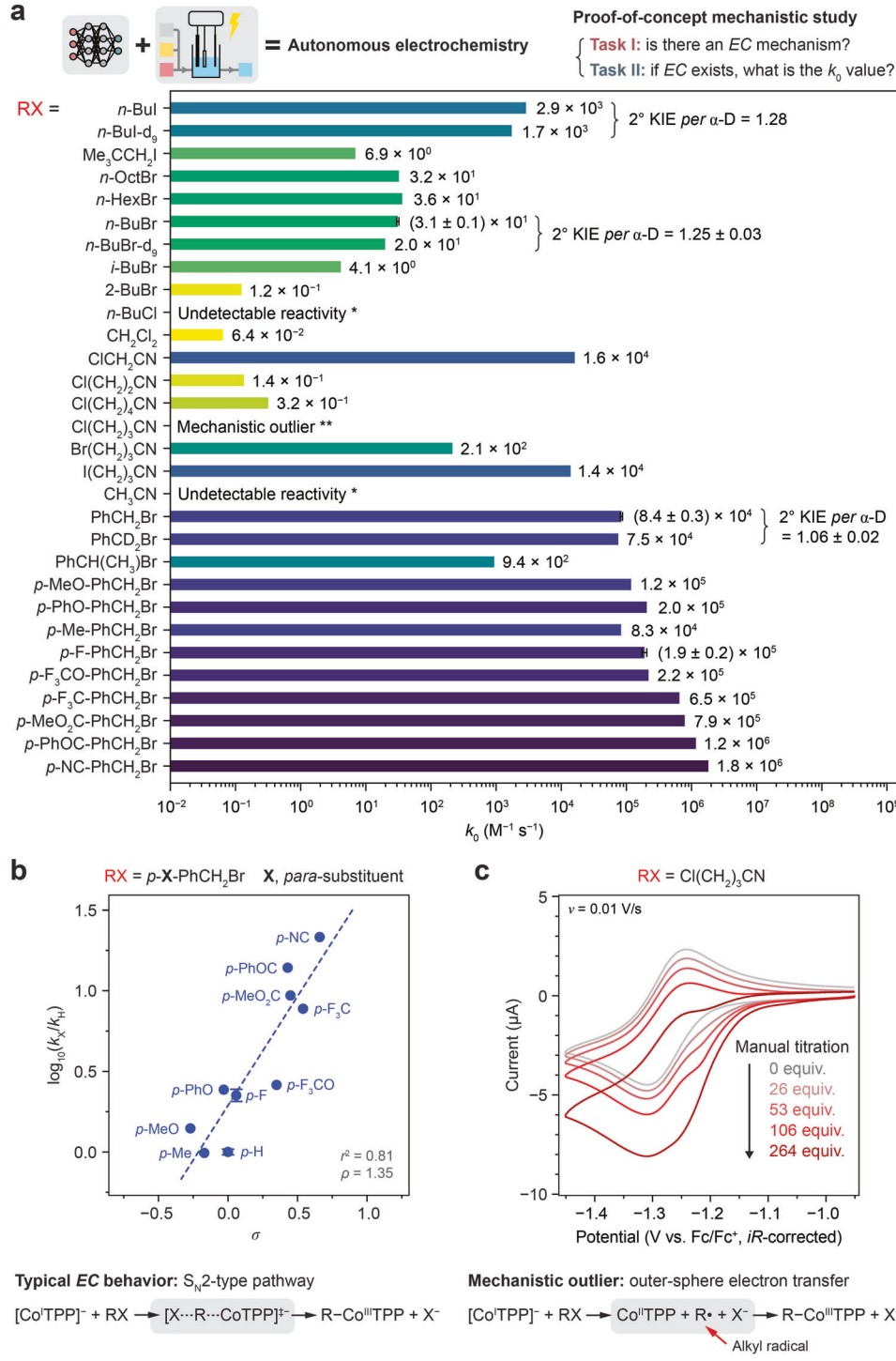

**Fig. 4 | Substrate scope studies and mechanistic insights. a** The generally applicable closed-loop workflow enables autonomous investigation of *EC* mechanism between CoTPP and a diverse scope of RX substrates amid negative controls and outliers, and the $k_0$ values of up to 30 substrates toward the electrogenerated Co$^I$TPP are measured over multiple orders of magnitude. * within the inquired parameter space; ** reacts with the electrogenerated Co$^I$TPP but does not follow an *EC* mechanism. **b** Hammett plot of *para*-substituted primary benzyl bromide derivatives on the $\sigma$ scale, where $k_X$ and $k_H$ are the $k_0$ values of the *para*-substituted (*p*-X-PhCH$_2$Br, denoted as *p*-X, where X represents a *para*-

substituent) and unsubstituted (PhCH$_2$Br, denoted as *p*-H) substrates, respectively. The error bars in (**a**) and (**b**) represent the standard deviations of $k_0$ and $\log_{10}(k_X/k_H)$, respectively, for those substrates measured by the closed-loop workflow for multiple replicates. **c** Additional manual CV experiments, via sequential titration of Cl(CH$_2$)$_3$CN with an increasing equivalent (equiv.) relative to Co$^{II}$TPP, confirm Cl(CH$_2$)$_3$CN as a mechanistic outlier that reacts with Co$^I$TPP but does not follow an *EC* mechanism. 1 mM Co$^{II}$TPP in DMF using 0.1 M NBu$_4$PF$_6$. Data distributions underlying the error bars in (**a**) and (**b**) are presented in Source Data file.

Supplementary Note 13) could be operational for Cl(CH$_2$)$_3$CN. As the DL model in the autonomous platform correctly identified the existence of alternative mechanisms, our platform is sensitive enough to detect outliers that bear previously unexpected mechanisms.

The performance of our demonstrated autonomous platform relies on the accuracy and general applicability of the DL-enabled automatic mechanism discernment. The DL model deployed here was trained by simulated voltammograms corresponding to only five

classical mechanisms and was tasked for mechanism classification with the assumption that there is only one redox event[34]. A more advanced DL model capable of both redox-event detection and mechanistic classification out of a larger library of nine mechanisms, dubbed as EchemNet, has recently been reported to accommodate voltammograms with in principle an arbitrary number of redox events[55]. Such advancement removes the restriction of pre-assigning a potential window for cyclic voltammetry measurements and will engender a higher level of autonomy in experimentation platforms. Nonetheless, advanced DL architectures are still needed to either account for mechanisms out of the existing trained ones or if possible accommodate all the possible mechanistic variations. The intrinsic issue of mechanistic ambiguity in voltammetry, that multiple mechanisms may exhibit similar if not the same voltammetric responses[35], suggests inevitable sampling bias during DL model's training. Thus, credible experimental data with proper mechanistic annotations should be welcomed to further refine the trained DL model and it is necessary to incorporate chemistry insights to further ascertain mechanism assignment.

## Discussion

In summary, we report an autonomous platform in fundamental electrochemistry that adaptively discerns molecular mechanisms and designs stringent experimental conditions for electrokinetic analysis. This modular platform is compatible with and readily applicable to existing electrochemistry laboratories, inheriting the rich knowledge in electrochemistry gained over the past decades. Through a proof-of-concept closed-loop survey of an EC mechanism between CoTPP and up to 30 RX substrates, we demonstrated the platform's wide range of rate quantification ($k_0$ of the C step) over 7 orders of magnitude, low relative uncertainty (<5%) for $k_0$ measurements, 10-fold reduction of experimental attempts compared to exhaustive screening, and the sensitivity and capability of detecting negative controls and mechanistic outliers. Such features will find great use for a myriad of mechanistic discoveries and investigations in molecular and organic electrochemistry. Future advances, such as ML models with expanded mechanistic capabilities of analyzing CV data and beyond as well as robotic handling of solid-state materials on electrodes, will lead to the establishment of general-purpose self-driving electrochemistry laboratories. The work presented here lays the cornerstone for such an edifice in the future.

## Methods

### Chemicals

Cobalt(II) tetraphenylporphyrin ($Co^{II}TPP$) (TCI America, >80.0%) was recrystallized from methylene chloride (Fisher Chemical, HPLC) before use: ~1 g of as-received $Co^{II}TPP$ was dissolved in ~500 ml of methylene chloride via sonication for ~2 h; the solution was filtered with 0.45 μm syringe filters (VWR), transferred in a crystallization dish (Pyrex), and evaporated under ambient atmosphere overnight; the recrystallized solid was dried at 60 °C and stored in an Argon-filled glovebox. Tetrabutylammonium hexafluorophosphate ($NBu_4PF_6$) (TCI America, >98.0%) was dried at 90 °C under vacuum and stored in the glovebox. Dimethylformamide (DMF) (Sigma-Aldrich, anhydrous, 99.8%) was stored in the glovebox and used as received.

A diverse scope of organic electrophiles was studied in this work. Purchased from Sigma-Aldrich were 1-bromobutane (n-BuBr) (99%), 1-iodobutane (n-BuI) (99%, stabilized with copper), 1-bromo-2-methylpropane (i-BuBr) (99%), dichloromethane ($CH_2Cl_2$) (anhydrous, ≥99.8%, stabilized with amylene), chloroacetonitrile ($ClCH_2CN$) (99%), acetonitrile ($CH_3CN$) (anhydrous, 99.8%), benzyl bromide-α,α-$d_2$ ($PhCD_2Br$) (98 atom% D), 4-methylbenzyl bromide (p-Me-$PhCH_2Br$) (97%), 4-(trifluoromethoxy)benzyl bromide (p-$F_3CO$-$PhCH_2Br$) (97%), methyl 4-(bromomethyl)benzoate (p-$MeO_2C$-$PhCH_2Br$) (98%), and 4-(bromomethyl)benzophenone (p-$PhOC$-$PhCH_2Br$) (96%). Purchased

from TCI America were 1-bromohexane (n-HexBr) (>98.0%), 1-bromooctane (n-OctBr) (>98.0%), 2-bromobutane (2-BuBr) (>98.0%), 1-chlorobutane (n-BuCl) (>99.0%), 3-chloropropionitrile ($Cl(CH_2)_2CN$) (>98.0%), 4-chlorobutyronitrile ($Cl(CH_2)_3CN$) (>97.0%), 5-chlorovaleronitrile ($Cl(CH_2)_4CN$) (>97.0%), 4-bromobutyronitrile ($Br(CH_2)_3CN$) (>97.0%), and benzyl bromide ($PhCH_2Br$) (>98.0%, stabilized with propylene oxide). Purchased from Oakwood Chemical were neopentyl iodide ($Me_3CCH_2I$) (98%), 4-iodobutyronitrile ($I(CH_2)_3CN$) (97%), and 1-(bromomethyl)-4-methoxybenzene (p-MeO-$PhCH_2Br$) (stabilized with $K_2CO_3$). Purchased from Synquest Laboratories were 4-fluorobenzyl bromide (p-F-$PhCH_2Br$) (98%), 4-(trifluoromethyl)benzyl bromide (p-$F_3C$-$PhCH_2Br$) (98%), and 4-cyanobenzyl bromide (p-NC-$PhCH_2Br$) (98%). Purchased from Cambridge Isotope Laboratories were n-BuBr-$d_9$ (98%) and n-BuI-$d_9$ (98%, stabilized with copper). Purchased from AK Scientific was 1-(bromomethyl)-4-phenoxybenzene (p-PhO-$PhCH_2Br$) (95%). Purchased from Thermo Scientific Chemicals was (1-bromoethyl)benzene ($PhCH(CH_3)Br$) (97%). Among all the organic electrophiles mentioned above, solid chemicals were stored in the glovebox and used as received, whereas liquid chemicals were transferred in a Schlenk flask, evacuated under vacuum on a Schlenk line, brought into the glovebox, and dried over 3 Å molecular sieves (Sigma-Aldrich) before use.

### Electrochemical experiments

All the stock solutions used for electrochemical experiments were prepared in the glovebox. Electrochemical experiments were performed in a single-compartment, three-electrode cell (placed in the glovebox) with a glassy carbon disk (3 mm in diameter) working electrode, a non-aqueous $Ag/Ag^+$ reference electrode (the filling solution was prepared as 10 mM $AgNO_3$ in anhydrous $CH_3CN$ using 0.1 M $NBu_4PF_6$ supporting electrolyte), and a platinum wire counter electrode connected to a CHI760E potentiostat (all from CH Instruments). The working electrode was polished with 0.05 μm Micro-Polish alumina powder on a MicroCloth polishing pad (both from CH Instruments) pre-wet with deionized water, thoroughly rinsed with deionized water, briefly sonicated in acetone for less than 20 s, and dried under ambient conditions before use. All measured potentials were versus the $Ag/Ag^+$ reference electrode, whose potential was calibrated against a ferrocene/ferrocenium ($Fc/Fc^+$) redox couple ($E$ versus $Ag/Ag^+$ = $E$ versus $Fc/Fc^+$ + 0.072 V) and remained fairly stable throughout this work (less than -10 mV drift over long-term storage). The formal potential for the $Co^{II/I}TPP$ redox was determined as −1.275 V versus $Fc/Fc^+$. Cyclic voltammetry was measured with automatic $iR$ compensation for three cycles, where the forward scan was a cathodic scan from −0.9 V vs. $Ag/Ag^+$ to the switching potential (set to −1.5 V vs. $Ag/Ag^+$ or a less negative value to avoid electroreduction of the metal−alkyl species formed from oxidative addition of organic electrophiles to the electrogenerated $Co^ITPP$), and the reverse scan was an anodic scan from the switching potential to −0.9 V vs. $Ag/Ag^+$. For the system of CoTPP with organic electrophiles studied in this work, electrode fouling was not observed for the working electrode during prolonged electrochemical testing (hundreds or thousands of cyclic voltammetry measurements during the autonomous closed-loop workflow or automated exhaustive screening, respectively), which allowed for continuous operation without the need of re-polishing the working electrode in the middle of each electrochemical experiment.

### Autonomous electrochemical platform

Detailed information about our constructed autonomous electrochemical platform is provided in the Supplementary Information, including but not limited to hardware and software specifications, standard operating procedures, and rationales behind our design of a closed-loop workflow for autonomous investigation of an EC mechanism between CoTPP and a library of organohalide substrates in

two stages (Stage I for mechanism discernment and Stage II for electrokinetic analysis).

## Data availability

The data that support the findings of this study are presented in the article and Supplementary Information. All new data associated with this paper are available in the Zenodo repository (https://doi.org/10.5281/zenodo.10587576). Any other relevant data are also available from the corresponding authors upon request. Source data are provided with this paper.

## Code availability

The source Python code is provided as Supplementary Software in this paper. The deep-learning model is available in the Zenodo repository (https://doi.org/10.5281/zenodo.10587576) due to large file sizes. Any other relevant code is also available from the corresponding authors upon request.

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

## Acknowledgements

Funding for this work was provided by the National Science Foundation (NSF) (CHE-2140762 and CHE-2247426 to C.L. and Q.G.; CHE-2102266 to A.G.D.; CHE-2002158 (Center for Synthetic Organic Electrochemistry) to M.S.S.), the Joint Center for Energy Storage Research (JCESR), an Energy Innovation Hub funded by the U.S. Department of Energy, Office of Science, Basic Energy Sciences (to J.R.L.), and partially the Agence Nationale de la Recherche (Labex ARCANE, CBH-EUR-GS, ANR-17-EURE-0003 to C.C.). H.S. acknowledges Dr. Shuangning Xu and Mr. Brandon Jolly for helpful discussions about chemical purification techniques.

## Author contributions

C.L. supervised the project. H.S. designed and built the experimental platform and performed all the experiments and mechanistic investigations. J.S. developed the computer codes for automated hardware control, Bayesian algorithms, and the autonomous closed-loop workflow. O.R. and J.R.-L. developed the Hard Potato Python library for automated electrochemical testing. B.B.H., W.Z., C.C., Q.G. and C.L. developed the deep-learning model for automated voltammogram analysis. D.X. performed proton nuclear magnetic resonance characterization and helped with the design of the experimental platform. T.T., A.H., D.S.M., A.G.D., M.S.S. and C.C. provided suggestions and feedback on the substrate scope studies and mechanistic investigations. H.S. wrote the initial draft of the manuscript. H.S. and C.L. revised and finalized the manuscript with the input from all authors.

## Competing interests

B.B.H., W.Z., Q.G., and C.L. are the inventors of a patent application (PCT/US2023/068008) for the deep-learning model used in this work. The remaining authors declare no competing interests.
