## [Peer Review File · Nature Communications]

Reviewers' Comments:

Reviewer #1:

Remarks to the Author:

Liu and coworkers presented a beautiful work on autonomous mechanistic investigations of cobalt tetraphenylporphyrin-mediated organohalide activation. They have carefully provided a significant amount of experimental details. Below are a few minor comments.

1. Did the authors observe working electrode faulting and Ag/Ag⁺ reference electrode leaching during the tens of hours-long experiment? If so, how did they determine and mitigate it?
2. The authors used two approaches for kinetic analysis: peak position shift and peak height ratio. They provided the workflow chart for the latter but not the former one. It would be useful if they could provide it.
3. I noticed some CVs in the SI indicates more complex mechanism than simple EC. For example, Figure 26d, the 3rd panel to the left in Figure 31a, the 3rd panel to the left in Figure 32a, Figure 33d, and the 2nd and 3rd panels to the left in Figure 34 all showed a cathodic prewave before the main peak. Some CVs will satisfy the ipa/ipc ratio between 0.65 and 0.75. How did the authors handle this situation?

Reviewer #2:

Remarks to the Author:

The manuscript "Autonomous closed-loop mechanistic investigation of molecular electrochemistry via automation" presents a novel approach in the field of electrochemistry, combining machine learning, flow chemistry, and electrochemical testing in a cohesive, automated platform. This study is particularly focused on exploring the reactivity of cobalt tetraphenylporphyrin with various electrophiles using an electrochemical (EC) mechanism. The integration of deep learning for data analysis and Bayesian optimization for experimental design is a groundbreaking advancement, offering new methodologies and insights into molecular electrochemistry.

The findings of this research are significant, as they demonstrate the platform's ability to efficiently identify reaction mechanisms and quantify kinetic rate constants. The paper and its supplementary information are well-structured, showcasing the experimental setup in detail, which includes both hardware and software specifications, along with standard operating procedures. This comprehensive detailing underscores the robustness of the methodology and its potential for reproducibility in further studies.

A notable aspect of this research is that it represents one of the first instances where the experimental outcome is mechanistic knowledge, rather than simply identifying conditions for a functional or synthetic yield. This shift in focus is significant as it achieves a substantial reduction in the time required to reach experimental goals, mirroring the impact typically seen in yield-focused research. Importantly, the authors have demonstrated a transferable methodology that effectively utilizes advanced data science and machine learning tools. This methodology can be applied broadly for extracting knowledge and understanding in various scientific domains, showcasing its versatility and potential for widespread application in research.

This manuscript represents a significant leap forward in integrating modern computational methods with traditional experimental techniques in electrochemistry. The approach is innovative, the findings are impactful, and the presentation of the work is both thorough and precise. The authors have commendably distributed the code associated with their research as well as the part and items for the build, significantly enhancing the potential for replication and further exploration by other researchers. This open approach to sharing their software tools is a strong indicator of the study's reliability and contributes to advancing the field as a whole. By making their code available, the authors not only enable other researchers to replicate their system and study but also provide a foundation for further advancements in similar projects. This openness and collaborative spirit are essential for the progressive evolution of both electrochemistry and robotics research.

Reviewer #3:

Remarks to the Author:

In this article by Sheng, Liu and co-workers, an autonomous workflow for the deciphering of mechanisms and rate constant in a typical molecular electrochemistry context is described. Molecular electrochemistry is a powerful tool, both for applications in green synthesis and energy related fields, as well as for the studying of reactions mechanisms of (electro)-catalytic systems. A short-coming, is the large parameter space that needs to be sampled in order to find suitable conditions and to deduce quantitative data. This work is thus highly relevant for the field, by combining ML-based analysis of experimental data with automated data generation and thus suitable for publication in Nature Communications. Before publications however, several points have to be addressed:

- early work from Speiser and colleagues on automated CV-analysis of organometallic species should be cited (e.g. 10.1016/j.elecom.2005.07.002)
- How are the 7% error in flow rate, as well as values below 100% for an EC mechanism propagated into k_0 ? How do you explain lower stdv in k_0 than the error in flow rate (and thus in concentration)?
- The CVs in Figure 2a are understood in terms of a EC mechanism with an irreversible C step and a large equilibrium constant. Are there other situations that could give similar results and thus false positive outputs? What is the influence of electron transfer kinetics and transfer coefficient on the validity of the mechanistic model?
- The same working electrode is used for a prolonged period of time. What is the effect of electrode fouling and change in capacitive current over time? In particular, how is I_{ps} (switching potential current) affected over time? Are interferences from other faradaic events (background reductions) observed and how are they treated?
- It seems that $k(\text{obs})$ does vary with i_{pa}/i_{pc} at 20mM, but not at lower values. What are the reasons of the "spikes" in the 20 mM curve (Figure 2C) and how is this taken into account?
- In the adaptive close-loop workflow, it seems like $[RX]$ values are probed for which the pseudo-first order approximation breaks down. How is this taken into account?
- The color coding legend in Fig. 3C/F is unclear in my opinion, it suggests that uncertainty of the GP model increases with additional data points. Why not express as in the SI as % ?
- The on the fly updated response surfaces don't seem to converge with additional data,, what does this imply for the GP model?
- When the peak potential is used to derive k_0 , the linear fits are often very bad, with roughly 30% error compared to the i_{pc}/i_{pa} technique and a 10% standard deviation. Is human intervention needed to assign the linear fit?

This work is a remarkable addition to the field, but any automated mechanistic analysis of reaction mechanisms or in general, the use of mechanistic models in electrochemistry can easily be used in cases where the models break or is not suitable. This is particularly true for autonomous workflows and ML models trained on specific subsets of (electro)-chemical reactivity. Reactions falling into the E/EC/CE/DISP etc subspace of electrochemical reactivity can be rare. In my opinion, a paragraph and critical discussion on the shortcomings of the model, limitations and underlying assumptions is thus indispensable in the main text of the manuscript.

We thank the reviewers for their very favorable reviews and approval of this manuscript and for their helpful comments and suggestions. Below, we provide detailed responses to the various reviewer concerns.

Reviewer #1

Liu and coworkers presented a beautiful work on autonomous mechanistic investigations of cobalt tetraphenylporphyrin-mediated organohalide activation. They have carefully provided a significant amount of experimental details. Below are a few minor comments.

We thank the reviewer for the favorable comments on our manuscript.

1. Did the authors observe working electrode fouling and Ag/Ag⁺ reference electrode leaching during the tens of hours-long experiment? If so, how did they determine and mitigate it?

We appreciate the reviewer's insights that the stability and reproducibility of working and reference electrodes are critical for automated experimentation over long durations. In short, both electrodes remain functional and stable during our automated experimentation.

(a) The working electrode.

Practically, we did not observe fouling of the glassy carbon working electrode within the duration of multi-hour operation. Our study has greatly benefited from the fact that the investigated electrochemical system undergoes fast quasi-reversible if not reversible *outer-sphere electron transfer that does not involve a strong interaction of the Co^{II/I}/TPP redox species with the working electrode*, which significantly reduces the possibility of undesired degradation of redox species on the working electrode. Besides, our results show that the follow-up chemical reaction (C step) of electrogenerated Co^ITPP species with the organohalide (RX) substrates studied in this work generally follow an *S_N2-type pathway that does not involve reactive radical intermediates*, which also reduces the chances of undesired side reactions on the working electrode compared to other possible radical-involving pathways (see Supplementary Note 11). A relevant discussion is also provided in our response to Comment #4 from Reviewer #3.

In the Methods section of the original manuscript, at the end of the paragraph on electrochemical experiments, we noted the absence of working electrode fouling for the investigated electrochemical system: *“For the system of CoTPP with organic electrophiles studied in this work, electrode fouling was not observed for the working electrode during prolonged electrochemical testing (hundreds or thousands of cyclic voltammetry measurements during the autonomous closed-loop workflow or automated exhaustive screening, respectively), which allowed for continuous operation without the need of re-polishing the working electrode in the middle of each electrochemical experiment.”*

(b) The Ag/Ag⁺ reference electrode.

While it is true the Ag⁺ species in the reference electrode can leach out to interfere with our experiments, in

practice we mitigated Ag/Ag⁺ reference electrode leaching by taking the following precautions: (i) every time before setting up a new experiment, the filling solution (10 mM AgNO₃ in anhydrous acetonitrile containing 0.1 M NBu₄PF₆) for the nonaqueous Ag/Ag⁺ reference electrode (CHI112 from CH Instruments) was replenished, with a stable liquid level achieved without noticeable dripping through the porous frit; (ii) when setting up the three-electrode cell, the porous frit of the reference electrode should not touch the bottom of the electrochemical cell; (iii) as the reference electrode was assembled in the glovebox under a slight overpressure above the atmospheric pressure, we set an overpressure of >5 mbar when the glovebox was idle so that the pressure difference between the reference electrode's inner headspace and the glovebox was effectively reduced if not eliminated. We found that both (ii) and (iii) were critical to mitigating Ag/Ag⁺ reference electrode leaching. Nevertheless, the use of a leak-free reference electrode (available from vendors such as Innovative Instruments, Inc.) could be attempted in the future.

2. The authors used two approaches for kinetic analysis: peak position shift and peak height ratio. They provided the workflow chart for the latter but not the former one. It would be useful if they could provide it.

We thank the reviewer for mentioning this. We note that, no matter which approach is used for kinetic analysis, the same autonomous workflow shown in Fig. 1f in the main text (as reproduced below) is used for experiment execution and data generation. The only difference lies in data analysis to extract k_0 , which is highlighted in the red dashed box in the figure below. For highly reactive benzyl bromide substrates studied in this work, there are too few voltammograms that satisfy $i_{pa}/i_{pc} \in [0.65, 0.75]$, occurring under at most one or two RX concentrations (see Supplementary Figs. 21–23, 25–34), which is insufficient for deriving the kinetic rate law as there are too few data points along the concentration axis (i.e., a meaningful linear fitting requires at least three data points).

Therefore, for highly reactive benzyl bromide substrates, upon the completion of all the experiment execution and data generation by the autonomous workflow, all the voltammograms measured at the lowest scan rate (0.01 V/s) in the parameter space from both Stage I and Stage II are selected for extracting k_0 from the shift of the cathodic peak potential (E_{pc}). Selecting the lowest scan rate for kinetic analysis using the E_{pc} approach is for the

sake of accuracy in measuring E_{pc} , as discussed in Supplementary Note 10. *In the revised resubmission, we have added a figure to show the revised workflow for kinetic analysis using the E_{pc} approach (as reproduced below) at the end of Supplementary Note 10.*

3. I noticed some CVs in the SI indicate more complex mechanisms than simple EC. For example, Figure 26d, the 3rd panel to the left in Figure 31a, the 3rd panel to the left in Figure 32a, Figure 33d, and the 2nd and 3rd panels to the left in Figure 34 all showed a cathodic prewave before the main peak. Some CVs will satisfy the i_{pa}/i_{pc} ratio between 0.65 and 0.75. How did the authors handle this situation?

We note that such complicated voltammetric responses (as mentioned by the reviewer) only happened to highly reactive *para*-substituted benzyl bromide substrates (p -X-PhCH₂Br, where X is a *para*-substituent) whose k_0 values are $\sim 10^5 \text{ M}^{-1} \text{ s}^{-1}$ or above (see Fig. 4a in the main text). We would like to clarify that such complicated voltammetric responses are not due to more complex mechanisms than a simple EC mechanism, but because *the pseudo-first-order condition for an EC mechanism (which is still the operating mechanism) no longer applies.*

To elaborate, as the organohalide (RX) substrate becomes highly reactive toward the electrogenerated Co^ITPP, on the one hand the RX substrate triggers a fast C step under a small RX concentration, resulting in *an anodic shift of the Co^{III} cathodic wave near the foot-of-the-wave region*, but on the other hand the local depletion of the RX substrate near the electrode surface is so pronounced near the foot-of-the-wave region that *the position of the Co^{III} cathodic wave beyond the foot-of-the-wave region remains largely unaffected*, resulting in a cathodic prewave before the main peak.

For these voltammograms, the spatiotemporal concentration of the RX substrate can no longer be considered as

constant and equal to its bulk concentration, and the pseudo-first-order condition assuming RX is in large excess for an *EC* mechanism no longer applies. Although some of these voltammograms satisfy $i_{pa}/i_{pc} \in [0.65, 0.75]$, they are not suitable for kinetic analysis of the *C* step using the i_{pa}/i_{pc} approach that assumes the pseudo-first-order condition. As shown in Supplementary Figs. 25–34, voltammograms that do not necessarily fulfill the pseudo-first-order condition but satisfy $i_{pa}/i_{pc} \in [0.65, 0.75]$ are quite rare for highly reactive *p*-X-PhCH₂Br substrates, occurring under at most one or two RX concentrations, which is insufficient for deriving the kinetic rate law as there are too few data points along the concentration axis (i.e., a meaningful linear fitting requires at least three data points). Therefore, *such voltammograms do not affect kinetic analysis of the C step for highly reactive p-X-PhCH₂Br substrates as the i_{pa}/i_{pc} approach cannot be used*, and an alternative approach based on the forward peak position (the E_{pc} approach, see Supplementary Note 10) is used instead (see the figure captions of Supplementary Figs. 25–34) based on voltammograms measured under other conditions that still fulfill the pseudo-first-order condition under sufficiently high RX concentrations (see Supplementary Figs. 35 and 36, also see Eq. S9 in Supplementary Note 10 for theoretical foundation).

Reviewer #2

The manuscript "Autonomous closed-loop mechanistic investigation of molecular electrochemistry via automation" presents a novel approach in the field of electrochemistry, combining machine learning, flow chemistry, and electrochemical testing in a cohesive, automated platform. This study is particularly focused on exploring the reactivity of cobalt tetraphenylporphyrin with various electrophiles using an electrochemical (EC) mechanism. The integration of deep learning for data analysis and Bayesian optimization for experimental design is a groundbreaking advancement, offering new methodologies and insights into molecular electrochemistry.

The findings of this research are significant, as they demonstrate the platform's ability to efficiently identify reaction mechanisms and quantify kinetic rate constants. The paper and its supplementary information are well-structured, showcasing the experimental setup in detail, which includes both hardware and software specifications, along with standard operating procedures. This comprehensive detailing underscores the robustness of the methodology and its potential for reproducibility in further studies.

A notable aspect of this research is that it represents one of the first instances where the experimental outcome is mechanistic knowledge, rather than simply identifying conditions for a functional or synthetic yield. This shift in focus is significant as it achieves a substantial reduction in the time required to reach experimental goals, mirroring the impact typically seen in yield-focused research. Importantly, the authors have demonstrated a transferable methodology that effectively utilizes advanced data science and machine learning tools. This methodology can be applied broadly for extracting knowledge and understanding in various scientific domains, showcasing its versatility and potential for widespread application in research.

This manuscript represents a significant leap forward in integrating modern computational methods with traditional experimental techniques in electrochemistry. The approach is innovative, the findings are impactful, and the presentation of the work is both thorough and precise. The authors have commendably distributed the code associated with their research as well as the part and items for the build, significantly enhancing the potential for replication and further exploration by other researchers. This open approach to sharing their software tools is a strong indicator of the study's reliability and contributes to advancing the field as a whole. By making their code available, the authors not only enable other researchers to replicate their system and study but also provide a foundation for further advancements in similar projects. This openness and collaborative spirit are essential for the progressive evolution of both electrochemistry and robotics research.

We thank the reviewer for the very favorable comments on and approval of our manuscript.

Reviewer #3

In this article by Sheng, Liu and co-workers, an autonomous workflow for the deciphering of mechanisms and rate constant in a typical molecular electrochemistry context is described. Molecular electrochemistry is a powerful tool, both for applications in green synthesis and energy related fields, as well as for the studying of reactions mechanisms of (electro)-catalytic systems. A short-coming, is the large parameter space that needs to be sampled in order to find suitable conditions and to deduce quantitative data. This work is thus highly relevant for the field, by combining ML-based analysis of experimental data with automated data generation and thus suitable for publication in Nature Communications. Before publication however, several points have to be addressed:

We thank the reviewer for the favorable comments on our manuscript.

1. Early work from Speiser and colleagues on automated CV-analysis of organometallic species should be cited (e.g. 10.1016/j.elecom.2005.07.002)

We thank the reviewer for bringing up this early work from Speiser and co-workers (*Electrochem. Commun.* **2005**, 7, 1013–1020), where cyclic voltammetric redox screening of ruthenium(II) complexes was conducted in a 96-well microliter plate, and their redox properties (the voltammogram shape or position) were further related to their catalytic activities for homogeneous hydrogenation reactions. Considering the experimental apparatus used in Speiser and co-workers' work, we found it appropriate to cite this work as ref. 19 in the second paragraph of the Introduction section, with the relevant context in the revised manuscript reproduced below:

“There have been key developments in high-throughput experimentation hardware for synthetic purposes¹²⁻¹⁷ that can be translated to electrochemical research¹⁸⁻²⁴, yet additional automated electroanalytical platforms with minimized adoption barriers are needed for fundamental mechanistic investigations. Customized multi-well parallel-plate reactors¹⁸⁻²² have been reported for evaluating organic electrosynthesis, and microfabricated microfluidic devices^{23,24} have been reported for automated electrokinetic measurements.”

2. How are the 7% error in flow rate, as well as values below 100% for an EC mechanism propagated into k_0 ? How do you explain lower stdv in k_0 than the error in flow rate (and thus in concentration)?

(a) How the deep-learning-generated mechanism propensity percentage propagates into k_0 .

Contrary to what the reviewer might think, the values below 100% for the deep-learning-generated propensity of an EC mechanism are not propagated into k_0 (the second-order kinetic rate constant of the C step) because the utility of the deep-learning model is solely in Stage I (mechanism discernment) but not in Stage II (electrokinetic analysis). As a result, k_0 is derived from i_{pa}/i_{pc} (reverse-to-forward peak current ratio) or E_{pc} (cathodic peak position) of voltammograms measured under suitable conditions and is unrelated to the deep-learning-generated propensity of an EC mechanism.

(b) Error propagation from the noted 7% error in flow rate.

In short, the noted 7% experimental error in flow rate will NOT significantly affect the measurement standard deviation of k_0 .

We assume that the 7% error in flow rate mentioned by the reviewer is sourced from the last figure in Section 1 of Supplementary Note 2 (as reproduced below): every time before setting up a new experiment, we routinely performed flow rate check for the three channels used for electrolyte formulation (Channel #1 flows $\text{Co}^{\text{II}}\text{TPP}$ stock solution; Channel #2 flows RX stock solution; Channel #3 flows NBu_4PF_6 supporting electrolyte solution), using anhydrous DMF solvent for flow rate check and ran each channel individually at 50 rpm for 30 seconds. The statistical results of 31 routine flow rate checks (see the right panel in the figure below) showed that the relative flow rates of these three channels were Channel #1 : Channel #2 : Channel #3 = 1 : (1.07 ± 0.03) : (1.02 ± 0.04) when running at the same rotation speed of 50 rpm. Such error in flow rate in principle is propagated into k_0 , but we did not consider such error propagation in our original submission.

To be scientifically rigorous, here we perform new analyses to consider the propagation of flow rate errors into k_0 . We made an assumption that the relative flow rates of the three channels of interest when running at the same rotation speed remain the same over the entire rotation speed range, otherwise it would be impossible to calculate error propagation. We revisited the three replicates of the autonomous investigation of $n\text{-BuBr}$ by the closed-loop workflow (see Supplementary Fig. 7) and reanalyzed the k_0 values by considering the propagation of flow rate errors. The results of routine flow rate check before setting up these three experiments are shown in the table below.

Experiment (see Supplementary Fig. 7)	Flow rate check before setting up the corresponding experiment
$n\text{-BuBr}$ (replicate #1)	Channel #1 : Channel #2 : Channel #3 = 1 : 1.05 : 0.95 (the 2nd flow rate check in the figure above)
$n\text{-BuBr}$ (replicate #2) ^[a]	Channel #1 : Channel #2 : Channel #3 = 1 : 1.06 : 1.02 (the 1st flow rate check in the figure above)
$n\text{-BuBr}$ (replicate #3) ^[a]	Channel #1 : Channel #2 : Channel #3 = 1 : 1.06 : 1.02 (the 1st flow rate check in the figure above)

^[a] Replicates #2 and #3 share the same flow rate check as these experiments were conducted on the same day.

Below we show the sample calculation of error propagation for the *n*-BuBr (replicate #1) experiment:

The absolute concentration of Co^{II}TPP does not affect kinetic analysis of the *C* step because k_{obs} is derived from the reverse-to-forward peak current *ratio* ($i_{\text{pa}}/i_{\text{pc}}$). However, the absolute concentration of RX would impact the derivation of k_0 from k_{obs} based on the rate law ($k_{\text{obs}} = k_0 [\text{RX}]^n$). Therefore, we only consider the error in the RX concentration in the error propagation calculation.

$$\text{Rate}_{\#1} = 50 \text{ rpm} \quad \text{Rate}_{\#2} \in [0, 50] \text{ rpm} \quad \text{Rate}_{\#2} + \text{Rate}_{\#3} = 50 \text{ rpm}$$

$$[\text{RX}]_{\text{targeted}} = [\text{RX}]_{\text{stock}} \times \frac{\text{Rate}_{\#2}}{\text{Rate}_{\#1} + \text{Rate}_{\#2} + \text{Rate}_{\#3}}$$

$$[\text{RX}]_{\text{actual}} = [\text{RX}]_{\text{stock}} \times \frac{1.05 \times \text{Rate}_{\#2}}{1 \times \text{Rate}_{\#1} + 1.05 \times \text{Rate}_{\#2} + 0.95 \times \text{Rate}_{\#3}}$$

$$\frac{[\text{RX}]_{\text{actual}}}{[\text{RX}]_{\text{targeted}}} = \frac{1.05 \times (\text{Rate}_{\#1} + \text{Rate}_{\#2} + \text{Rate}_{\#3})}{1 \times \text{Rate}_{\#1} + 1.05 \times \text{Rate}_{\#2} + 0.95 \times \text{Rate}_{\#3}}$$

Suitable data for kinetic analysis	$[\text{RX}]_{\text{targeted}}$	$[\text{RX}]_{\text{stock}}$	Rate _{#2}	$[\text{RX}]_{\text{actual}}$
Stage I (see Supplementary Fig. 6)	29.0 mM	2000 mM	1.45 rpm	31.2 mM
	164 mM	2000 mM	8.18 rpm	175 mM
Stage II (see Supplementary Fig. 6)	0.82 mM	40 mM	2.05 rpm	0.88 mM
	1.00 mM	40 mM	2.51 rpm	1.08 mM
	1.80 mM	40 mM	4.50 rpm	1.93 mM
	2.56 mM	40 mM	6.39 rpm	2.73 mM
	3.78 mM	40 mM	9.44 rpm	4.03 mM
	4.86 mM	40 mM	12.16 rpm	5.17 mM
	9.90 mM	40 mM	24.76 rpm	10.4 mM
	11.8 mM	40 mM	29.48 rpm	12.3 mM
24.0 mM	2000 mM	1.20 rpm	25.8 mM	

Before considering the propagation of flow rate errors, the rate law ($k_{\text{obs}} = k_0 [\text{RX}]^n$) is derived from $[\text{RX}]_{\text{targeted}}$ values shown in the table above, resulting in $k_0 = 31.2 \text{ M}^{-1} \text{ s}^{-1}$ (see the left panel of the figure below, which is the same as Supplementary Fig. 7g in the SI). *After considering the propagation of flow rate errors, the rate law is derived from $[\text{RX}]_{\text{actual}}$ values shown in the table above, resulting in $k_0 = 29.2 \text{ M}^{-1} \text{ s}^{-1}$ (see the right panel of the figure below).*

n-BuBr (replicate #1)
Before considering the propagation of flow rate errors

n-BuBr (replicate #1)
After considering the propagation of flow rate errors

Following the same logic detailed in the sample calculation above, we further considered the propagation of flow rate errors for the *n*-BuBr (replicate #2) and *n*-BuBr (replicate #3) experiments, with the k_0 values shown in the figure below.

n-BuBr (replicate #2)
Before considering the propagation of flow rate errors

n-BuBr (replicate #2)
After considering the propagation of flow rate errors

n-BuBr (replicate #3)
Before considering the propagation of flow rate errors

n-BuBr (replicate #3)
After considering the propagation of flow rate errors

From the three replicate experiments of *n*-BuBr, $k_0 = 30.9 \pm 1.3 \text{ M}^{-1} \text{ s}^{-1}$ before considering the propagation of flow rate errors, and $k_0 = 29.3 \pm 1.3 \text{ M}^{-1} \text{ s}^{-1}$ after considering the propagation of flow rate errors. *As shown in the table below, the relative standard deviation of k_0 remains almost the same after considering the propagation of flow rate errors.*

Experiment (see Supplementary Fig. 7)	$k_0 \text{ (M}^{-1} \text{ s}^{-1})$ before considering the propagation of flow rate errors ^[a]	$k_0 \text{ (M}^{-1} \text{ s}^{-1})$ after considering the propagation of flow rate errors
n -BuBr (replicate #1)	31.2	29.2
n -BuBr (replicate #2)	29.5	28.1
n -BuBr (replicate #3)	32.0	30.6
Mean ± standard deviation	30.9 ± 1.3	29.3 ± 1.3
Relative standard deviation	4.1%	4.3%

^[a] The k_0 values before considering the propagation of flow rate errors shown here are the same as those shown in Supplementary Fig. 7.

To conclude, we have performed detailed new analyses to show that *the propagation of flow rate errors into k_0 is a rather complicated process that does not necessarily amplify the relative standard deviation of k_0 beyond the relative error in flow rate.* Our results show that considering the propagation of flow rate errors barely affect the mean, standard deviation, and relative standard deviation of k_0 from three replicate experiments of *n*-BuBr.

3. *The CVs in Figure 2a are understood in terms of an EC mechanism with an irreversible C step and a large equilibrium constant. Are there other situations that could give similar results and thus false positive outputs? What is the influence of electron transfer kinetics and transfer coefficient on the validity of the mechanistic model?*

(a) Whether there are other situations that could give similar results to an EC mechanism.

We agree with the reviewer that it is possible that multiple mechanisms may lead to the similar and sometimes even the same voltammograms. But such mechanistic ambiguity originates from the technique of cyclic voltammogram per se, not from the developed deep-learning model deployed in this study.

In cyclic voltammetry, mechanistic ambiguity denotes the fact that different reaction mechanisms may yield similar voltammograms that are indistinguishable within measurement errors. For an EC mechanism with an irreversible C step bearing a large equilibrium constant, competition between chemical reaction and diffusion is governed by a dimensionless kinetic parameter $\lambda = \frac{RTk}{Fv}$ (where k is the rate constant of the C step, v is the scan rate). As λ increases, the voltammetric response becomes more irreversible. When λ is large enough to lead to a completely irreversible wave and reach *pure kinetic conditions*, the forward peak characteristics (peak current density i_p , peak potential E_p , and peak width $E_{p/2} - E_p$) are as follows (see Chapter 2 of *Elements of Molecular and Biomolecular Electrochemistry: An Electrochemical Approach to Electron Transfer Chemistry*, 2nd edition, by Savéant and Costentin):

$$i_p = 0.496FC^0\sqrt{D}\sqrt{\frac{Fv}{RT}} \quad E_p = E^0 - 0.78\frac{RT}{F} + \frac{RT}{2F}\ln\left(\frac{RTk}{Fv}\right) \quad E_{p/2} - E_p = 1.857\frac{RT}{F}$$

where E^0 , C^0 , and D are the standard potential, bulk concentration, and diffusion coefficient of the redox couple O/R (assuming $D = D_O = D_R$).

For *ECE* and *DISPI* mechanisms, when *pure kinetic conditions* are reached, the forward peak characteristics highly resemble those of an *EC* mechanism under *pure kinetic conditions*, with the same peak potential and peak width except for the peak current density being exactly twice (see Chapter 2 of *Elements of Molecular and Biomolecular Electrochemistry: An Electrochemical Approach to Electron Transfer Chemistry*, 2nd edition, by Savéant and Costentin):

$$i_p = 0.992FC^0\sqrt{D}\sqrt{\frac{Fv}{RT}} \quad E_p = E^0 - 0.78\frac{RT}{F} + \frac{RT}{2F}\ln\left(\frac{RTk}{Fv}\right) \quad E_{p/2} - E_p = 1.857\frac{RT}{F}$$

That said, without a priori knowledge of E^0 , C^0 , and D of the redox couple O/R, the voltammetric responses of *EC*, *ECE*, and *DISPI* mechanisms under *pure kinetic conditions* are indistinguishable (i.e., the same scan rate dependence of $i_p \propto \sqrt{v}$, and the same peak potential and peak width). In principle, mechanistic designation is impossible for this scenario of mechanistic ambiguity, regardless of whether analysis is manual-based or deep-learning-based.

We then went through our experimental data to find cases of an *EC* mechanism under *pure kinetic conditions* and the corresponding deep-learning-generated mechanism propensity distributions. Our deep-learning model occasionally yields a high propensity value of *ECE* mechanism for voltammograms of an *EC* mechanism under *pure kinetic conditions*, as exemplified by highlighted cases in dashed red boxes in the figure below (these data with RX = *n*-HexBr are reproduced from panels (a) and (c) of Supplementary Fig. 10, with panels (a₆), (a₈), (a₁₄) and their corresponding entries in panel (c) highlighted). For most cases of an *EC* mechanism under *pure kinetic conditions*, the yielded propensity values of an *ECE* or *DISPI* mechanism are zero or marginal and do not affect the assignment of an *EC* mechanism as the most probable mechanism (see Supplementary Figs. 3, 4, 8–23, 25–34 for the entire substrate scope studied in this work. Therefore, our deep-learning model seems to exhibit a bias toward assigning an *EC* mechanism when *EC*, *ECE*, and *DISPI* mechanisms under *pure kinetic conditions* are in principle indistinguishable, and this bias is unintentional and could be related to any sampling bias in the training set or the model training process. Because mechanistic ambiguity is intrinsic of cyclic voltammetry, it is difficult to remove such a model bias without compromising practical utility of the deep-learning model.

(b) Effects of electron transfer kinetics and transfer coefficients.

In short, the deployed deep-learning model has already considered the effect of electron transfer kinetics while the model assumes a fixed transfer coefficient ($\alpha = 0.5$).

The deep-learning model in this manuscript was originally published in our previous work (*ACS Meas. Sci. Au* **2022**, *2*, 595–604), and concentration-dependent Butler–Volmer equation (Eq. R1) was employed to define the kinetics of single-electron transfer step(s) in each mechanism type (see details in the Supporting Information of *ACS Meas. Sci. Au* **2022**, *2*, 595–604):

$$i(t) = i_0 \left\{ \frac{[R]_{x=0}}{0.5 \cdot C_{R,i}} \exp \left[\frac{\alpha F}{RT} (E - E_{O/R}) \right] - \frac{[O]_{x=0}}{0.5 \cdot C_{R,i}} \exp \left[-\frac{(1-\alpha)F}{RT} (E - E_{O/R}) \right] \right\} \quad (\text{Eq. R1})$$

where $0.5 \cdot C_{R,i}$ denotes the equilibrium concentration when $E = E_{O/R}$ and $C_O = C_R$, and the transfer coefficient is assumed to be constant ($\alpha = 0.5$).

For each mechanism type, a population of >3000 simulated cyclic voltammograms was generated as the training set by sampling a wide range for the dimensionless quasi-reversibility parameter ($\psi \in [10, 0.3]$) following the Nicholson's formalism (Eq. R2):

$$i_0 = k_S \cdot 0.5FC_{R,i} = \psi \sqrt{\frac{\pi F \nu D}{RT}} \cdot 0.5FC_{R,i} \quad (\text{Eq. R2, assuming } D_O = D_R = D)$$

In the context of a quasireversible E step, $\psi \in [10, 0.3]$ corresponds to a peak separation of $62 \sim 120$ mV in the cyclic voltammograms: the upper bound of ψ is high enough that the resultant scenarios resemble the Nernstian scenario in cyclic voltammetry in which the interfacial charge transfer is fast enough to ensure a Nernstian equilibrium for the redox species in the immediate proximity near the electrode.

Here the transfer coefficient is assumed to be a constant value ($\alpha = 0.5$) for the following reasons (see page 132 of *Electrochemical Methods: Fundamentals and Applications*, 3rd edition, by Bard, Faulkner, and White): (1) in most experimental cases, α turns out to lie between 0.3 and 0.7, and is often taken to be 0.5 in the absence of actual measurements; (2) the Butler–Volmer kinetic model assumes α to be independent of potential, which appears to be true in most experiments because the potential range over which kinetic data can be collected is often fairly narrow.

To conclude, electron transfer kinetics defined by concentration-dependent Butler–Volmer equation have been considered in our deep-learning model, under the assumption of a potential-independent transfer coefficient of 0.5. Future improvements to our deep-learning model can be made by considering the transfer coefficient as a potential-independent variable ($0 < \alpha < 1$) or by considering the potential dependence of the transfer coefficient according to the Marcus theory (see page 152 of *Electrochemical Methods: Fundamentals and Applications*, 3rd edition, by Bard, Faulkner, and White).

4. The same working electrode is used for a prolonged period of time. What is the effect of electrode fouling and change in capacitive current over time? In particular, how is I_{ps} (switching potential current) affected over time? Are interferences from other faradaic events (background reductions) observed and how are they treated?

(a) The potential issue of electrode fouling and change in capacitive current over time.

Our data suggested that, at least for the chemical system studied in this work (the reactions of CoTPP with RX electrophiles), the working electrode did not experience noticeable electrode fouling or change in capacitive current over time during long-term operation of cyclic voltammetry. As double-layer capacitance measurements in the absence of redox-active species were not included in our experimental design of either the automated exhaustive experiments or the autonomous closed-loop workflow, the best case for us to examine the effect of electrode fouling and change in capacitive current over time based on our existing data was *to compare the voltammograms of CoTPP (in the presence or absence of RX) measured at the same scan rate toward the beginning versus the end of long-term experiments*.

We found it more feasible to find such comparable data from the automated exhaustive experiments because the voltammogram sets (recorded at six logarithmically-sampled scan rates, $v_{\min}/v_{\max} = 1/10$) were systematically measured for every designated combination of $[RX] \in [0, 20]$ mM (step size = 1 mM, $RX = n\text{-BuBr}$) and $v_{\min} \in [0.01, 0.2]$ V/s (step size = 0.01 V/s) in the search space, whereas the autonomous closed-loop workflow had more flexible assignments of $[RX]$ and v_{\min} that were not necessarily comparable over time.

To better visualize capacitive current, we selected the voltammograms measured at the maximum scan rate (2

V/s) in the search space for comparison. Since the ~50-hour automated exhaustive experiments surveyed the designated [*n*-BuBr] values incrementally, 0 mM *n*-BuBr was tested toward the beginning, and 20 mM *n*-BuBr was tested toward the end. The figure below shows the voltammograms of 1 mM Co^{II}TPP with 0 mM versus 20 mM *n*-BuBr, both measured at 2 V/s. The scan rate was fast enough so that the voltammetric response appeared to be quasi-reversible for both cases. These two voltammograms recorded over a timespan of ~50 hours were almost overlapping, with very similar redox peak currents and peak separations and without emerging features over time. Therefore, we concluded that the working electrode did not experience noticeable electrode fouling or change in capacitive current over time during the long-term operation of cyclic voltammetry.

(b) How is the switching potential current affected over time.

Speaking of how the switching potential current (i_{ps}) was affected over time, we again used the same set of selected data shown above for illustrative purposes. For the ~50-hour automated exhaustive experiments with *n*-BuBr, at the switching potential (-1.5 V vs. Ag/Ag⁺), the voltammogram measured at 2 V/s under 0 mM *n*-BuBr toward the beginning had an i_{ps} value of -2.66×10^{-5} A, and that under 20 mM *n*-BuBr toward the end had an i_{ps} value of -2.94×10^{-5} A. Such a small difference in the i_{ps} values of these two voltammograms recorded over a timespan of ~50 hours let us conclude that i_{ps} was not particularly affected over time.

(c) Whether there are interferences from other faradaic events.

In short, we contend that there is minimal interference from the other Faradaic events. The noted other faradaic events depend on the type and concentration of RX substrates. For example, for ClCH₂CN substrate that reacts with the electrogenerated Co^ITPP at a fast rate ($k_0 = 1.6 \times 10^4$ M⁻¹ s⁻¹, see Supplementary Fig. 16), when implementing the autonomous closed-loop workflow, the switching potential was set to a less negative potential of -1.35 V vs. Ag/Ag⁺, which was around the onset potential for the electroreduction of R-Co^{III}TPP (the product of the C step between Co^ITPP and RX) when $R = -CH_2CN$. This modified setting of the switching potential for the autonomous investigation of ClCH₂CN substrate was made on purpose to avoid the undesired faradaic event of R-Co^{III}TPP reduction, and this decision was made beforehand via manual recording of exemplary voltammograms of CoTPP in the presence of ClCH₂CN (see the figure below). For the other RX substrates studied in this work, the switching potential was set as -1.4 V vs. Ag/Ag⁺ for the autonomous closed-loop workflow.

For highly reactive *para*-substituted benzyl bromide substrates (*p*-X-PhCH₂Br, where X is a *para*-substituent) whose k_0 values are $\sim 10^5 \text{ M}^{-1} \text{ s}^{-1}$ or above (see Fig. 4a in the main text), we occasionally observed undesired faradaic current of R-Co^{III}TPP reduction in the measured voltammograms when the switching potential was set as $-1.4 \text{ V vs. Ag/Ag}^+$ for the autonomous closed-loop workflow (see exemplary voltammograms for *p*-MeO₂C-PhCH₂Br, *p*-PhOC-PhCH₂Br, and *p*-NC-PhCH₂Br substrates in Supplementary Fig. 35), affecting the i_{ps} values for these voltammograms. Nevertheless, as mentioned earlier in our response to Comment #3 from Reviewer #1, for highly reactive *p*-X-PhCH₂Br substrates, the i_{pa}/i_{pc} approach (which relies on i_{ps} for calculation) cannot be used for kinetic analysis of the C step due to the breakdown of the pseudo-first-order approximation, and an alternative approach based on the forward peak position (the E_{pc} approach, see Supplementary Note 10) is used instead and does not rely on i_{ps} for calculation (see Supplementary Figs. 35 and 36). Therefore, the occasionally observed faradaic current of R-Co^{III}TPP reduction for highly reactive *p*-X-PhCH₂Br substrates did not affect kinetic analysis of the C step.

In the original submission, we have mentioned the choice of the switching potential and its implication on other faradaic events in the Methods section: “Cyclic voltammetry was measured with automatic iR compensation for three cycles, where the forward scan was a cathodic scan from $-0.9 \text{ V vs. Ag/Ag}^+$ to the switching potential (set to $-1.5 \text{ V vs. Ag/Ag}^+$ or a less negative value to avoid electroreduction of the metal-alkyl species formed from oxidative addition of organic electrophiles to the electrogenerated Co^ITPP), and the reverse scan was an anodic scan from the switching potential to $-0.9 \text{ V vs. Ag/Ag}^+$.”

5. It seems that k_{obs} does vary with i_{pa}/i_{pc} at 20 mM, but not at lower values. What are the reasons of the “spikes” in the 20 mM curve (Figure 2C) and how is this taken into account?

We thank the reviewer for this meticulous observation. Here we plot all the voltammograms measured under 20 mM *n*-BuBr and different scan rates that satisfy $i_{pa}/i_{pc} \in [0.65, 0.75]$, along with the corresponding i_{pa}/i_{pc} value and the calculated k_{obs} value of the C step from each voltammogram, as shown in the figure below. We confirm that all these voltammograms (contributing to the 20 mM curve in Fig. 2c in the main text) display typical

voltammetric responses corresponding to an *EC* mechanism, without noticeable deviation or additional features. The calculated k_{obs} value under 20 mM *n*-BuBr was $0.417 \pm 0.022 \text{ s}^{-1}$ (mean \pm standard deviation), and the relative standard deviation (RSD) was 5.3%.

To compare the statistical noises for the calculated k_{obs} value under 20 mM *n*-BuBr versus other lower *n*-BuBr concentrations, we further plot all the voltammograms measured under 10 mM *n*-BuBr and different scan rates that satisfy $i_{\text{pa}}/i_{\text{pc}} \in [0.65, 0.75]$, along with the corresponding $i_{\text{pa}}/i_{\text{pc}}$ value and the calculated k_{obs} value of the C step from each voltammogram, as shown in the figure below. We found that the RSD of the calculated k_{obs} value under 10 mM *n*-BuBr was 4.0%, in fact very close to that under 20 mM *n*-BuBr (5.3% as calculated above).

To take into account the statistical noises for the calculated k_{obs} values under each *n*-BuBr concentration, when plotting $\log_{10}(k_{\text{obs}})$ versus $\log_{10}[n\text{-BuBr}]$ in Fig. 2e in the main text, we showed the mean (represented by dot) \pm standard deviation (represented by vertical error bar) of $\log_{10}(k_{\text{obs}})$ at each $\log_{10}[n\text{-BuBr}]$ value, based on all the k_{obs} values calculated from voltammograms that satisfy $i_{\text{pa}}/i_{\text{pc}} \in [0.65, 0.75]$ under each *n*-BuBr concentration.

6. In the adaptive close-loop workflow, it seems like $[RX]$ values are probed for which the pseudo-first order approximation breaks down. How is this taken into account?

This question is related to Comment #3 from Reviewer #1. We note that the breakdown of the pseudo-first order approximation only occurs when trying to use the i_{pa}/i_{pc} approach for kinetic analysis of the C step for highly reactive *para*-substituted benzyl bromide substrates (p -X-PhCH₂Br, where X is a *para*-substituent) whose k_0 values are $\sim 10^5$ M⁻¹ s⁻¹ or above (see Fig. 4a in the main text). The underlying reason for such a breakdown has been elaborated in our response to Comment #3 from Reviewer #1.

As noted in the figure captions of Supplementary Figs. 25–34, for highly reactive p -X-PhCH₂Br substrates, an alternative approach based on the forward peak position (the E_{pc} approach, see Supplementary Fig. 10) is used for kinetic analysis of the C step, which is based on voltammograms measured under other conditions that still fulfill the pseudo-first-order approximation under sufficiently high RX concentrations (see Supplementary Figs. 35 and 36, also see Eq. S9 in Supplementary Note 10 for theoretical foundation).

In short, the pseudo-first-order approximation always needs to be fulfilled for meaningful kinetic analysis of the C step in an EC mechanism. Since voltammograms that satisfy $i_{pa}/i_{pc} \in [0.65, 0.75]$ do not necessarily fulfill the pseudo-first-order approximation for highly reactive p -X-PhCH₂Br substrates and negate the utility of the i_{pa}/i_{pc} approach, an alternative E_{pc} approach is used instead meanwhile the voltammograms used for kinetic analysis still fulfill the pseudo-first-order approximation.

7. The color coding legend in Fig. 3C/F is unclear in my opinion, it suggest that uncertainty of the GP model increases with additional data points. Why not express as in the SI as % ?

(a) The color coding legend in Fig. 3c/f, and why certainty increases with additional data points.

We acknowledge the reviewer's observation that uncertainty of the Gaussian process results shown in Fig. 3c/f in the main text (as reproduced in the figure below) increases with additional data points in certain regions of the parameter space.

We would like to justify the use of the color coding legend in these figures as follows. These Gaussian process results were obtained by regressing the Gaussian process model to the i_{pa}/i_{pc} values of all the measured

voltammograms in the parameter space of $v \in [0.01, 2]$ V/s and $\log_{10}[n\text{-BuBr}] \in [\log_{10}(0.008 \text{ mM}), \log_{10}(1000 \text{ mM})]$. As i_{pa}/i_{pc} is a value between 0 and 1, the plotted i_{pa}/i_{pc} uncertainty is an absolute uncertainty (i.e., the standard deviation of the modelled i_{pa}/i_{pc}) rather than a relative uncertainty. To visualize the change in i_{pa}/i_{pc} uncertainty with additional data points, below we plot the differential i_{pa}/i_{pc} uncertainty in the parameter space of v and $\log_{10}[\text{RX}]$ as follows:

$$\text{differential } i_{pa}/i_{pc} \text{ uncertainty} = (i_{pa}/i_{pc} \text{ uncertainty after Stage II}) - (i_{pa}/i_{pc} \text{ uncertainty after Stage I})$$

We acknowledge that there is only a small region in the parameter space where i_{pa}/i_{pc} uncertainty decreases with additional data points, as indicated by the blue region in the differential i_{pa}/i_{pc} uncertainty plot above. The reason is in fact straightforward to rationalize: *the region where i_{pa}/i_{pc} uncertainty decreases coincides reasonably well with the region where additional data points are recorded* (the recorded data points from Stage I versus Stage II are visualized in the parameter space as black versus red points in the figure below).

Therefore, we consider the use of the color coding legend in Figs. 3c/f in the main text is reasonable, and our new analysis suggests the decrease of i_{pa}/i_{pc} uncertainty only occurs in a small region of the parameter space where additional data points are recorded but not the entire parameter space.

(b) Why uncertainty in Fig. 3c/f is not expressed as %.

We would like to draw the reviewer's attention to the difference between the Gaussian process results shown in Fig. 3c/f in the main text and those shown in Supplementary Note 7 in the SI. In Supplementary Note 7, the

Gaussian process is employed to model a response surface of *the deep-learning-generated propensity of an EC mechanism, which is intrinsically a percentage value between 0% to 100%*. That said, although the color coding legend is labeled as % for the local model uncertainty, this uncertainty is *the standard deviation of a percentage value and thus is still an absolute uncertainty* rather than a relative uncertainty.

8. *The on the fly updated response surfaces don't seem to converge with additional data, what does this imply for the GP model?*

We agree with the reviewer that, speaking of on-the-fly update of the response surface of i_{pa}/i_{pc} during Stage II of the closed-loop workflow when $RX = n\text{-BuBr}$, the response surfaces do not seem to converge with additional data (as shown in Supplementary Fig. 5). *The possible reason is that a small number of data points were fitted to a large parameter space.* The Gaussian process was performed in a large parameter space of $v \in [0.01, 2]$ V/s and $\log_{10}[RX] \in [\log_{10}(0.008 \text{ mM}), \log_{10}(1000 \text{ mM})]$. Right after Stage I (Bayesian optimization to maximize the voltammograms' propensity toward an EC mechanism), there were 15 (measured voltammogram sets) \times 6 (logarithmically-sampled scan rates per voltammogram set) = 90 data points of i_{pa}/i_{pc} (one per voltammogram). After 19 inquiries in Stage II (where each inquiry measured a voltammogram set under six proximate scan rates to aim for $i_{pa}/i_{pc} \in [0.65, 0.75]$), the number of data points of i_{pa}/i_{pc} was augmented to $(90 + 6 \times 19) = 204$. *The distribution of data points in the parameter space, both before and after the data augmentation in Stage II, was visualized in our response to Comment #7 from Reviewer #3.* Because of the large parameter space, the decrease in the uncertainty of the Gaussian process model only occurred in the small region of the parameter space where additional data points were augmented but not the entire parameter space, resulting in the non-convergence of the Gaussian process.

In the revised resubmission, we added a comment on the non-convergence of the Gaussian process in the figure caption of Supplementary Fig. 5: “The fluctuations in the surfaces and *the non-convergence of the Gaussian process with additional data points* could be ascribed to the fact that a small number of data points are fitted to a large parameter space.”

9. *When the peak potential is used to derive k_0 , the linear fits are often very bad, with roughly 30% error compared to the i_{pa}/i_{pc} technique and a 10% standard deviation. Is human intervention needed to assign the linear fit?*

(a) Response to “when the peak potential is used to derive k_0 , the linear fits are often very bad”.

We would like to provide some clarification regarding the reviewer’s statement that “when the peak potential is used to derive k_0 , the linear fits are often very bad”. As detailed in Supplementary Note 10, the theoretical foundation for kinetic analysis of the C step using the cathodic peak potential (E_{pc}) approach is as follows:

$$E_{pc} = E_{1/2} - 0.78 \frac{RT}{F} + \frac{RT \ln(10)}{2F} \log_{10} \left(\frac{RT k_0 [RX]}{Fv} \right) \quad (\text{Eq. R2})$$

where R is the ideal gas constant ($8.314 \text{ J mol}^{-1} \text{ K}^{-1}$), T is the temperature (K), F is the Faraday’s constant

(96485 C mol^{-1}), $E_{1/2}$ is the formal potential of the $\text{Co}^{\text{II/I}}$ redox, k_0 is the second-order kinetic rate constant ($\text{M}^{-1} \text{ s}^{-1}$) of the C step between $\text{Co}^{\text{I}}\text{TPP}$ and RX under a pseudo-first-order condition when RX is in large excess, [RX] is the RX concentration, and v is the scan rate.

Although Eq. R2 defines a linear relationship of E_{pc} versus $\log_{10}[\text{RX}]$ regardless of the [RX] value, *this linear relationship is only applicable to sufficiently high [RX] values that can cause a noticeable anodic shift of E_{pc} .* When [RX] = 0, E_{pc} in principle should be the theoretical cathodic peak potential for a reversible single-electron redox event ($E_{\text{pc}} = E_{1/2} - 1.11 \frac{RT}{F}$ for a reversible E step with fast kinetics, see page 7 of *Elements of Molecular and Biomolecular Electrochemistry: An Electrochemical Approach to Electron Transfer Chemistry*, 2nd edition, by Savéant and Costentin); *however, E_{pc} defined by Eq. R2 approaches $-\infty$ when [RX] approaches 0, making Eq. R2 inapplicable to the small [RX] regime.* Therefore, experimental data of E_{pc} remain constant in the small [RX] regime and display a linear increase with $\log_{10}[\text{RX}]$ only under sufficiently high [RX] values.

(b) Response to “roughly 30% error compared to the $i_{\text{pa}}/i_{\text{pc}}$ technique and a 10% standard deviation”.

We assume the ~30% error and ~10% standard deviation (as stated by the reviewer) for k_0 derived from the E_{pc} approach are sourced from Supplementary Note 10, and we would like to provide some clarification. First, the ~30% error is the *systematic error* between the two methodologies (the E_{pc} approach vs. the $i_{\text{pa}}/i_{\text{pc}}$ approach) of deriving k_0 , as exemplified by six representative RX substrates (*n*-BuI, *n*-BuI- d_9 , ClCH_2CN , $\text{Br}(\text{CH}_2)_3\text{CN}$, $\text{I}(\text{CH}_2)_3\text{CN}$, and $\text{PhCH}(\text{CH}_3)\text{Br}$) whose k_0 derived from the E_{pc} approach is *consistently* ~30% smaller than that derived from the $i_{\text{pa}}/i_{\text{pc}}$ approach. Second, the ~10% standard deviation serves as a *measure of consistency in the systematic error between the two methodologies across different RX substrates*. If the systematic error between the two methodologies is consistent, k_0 derived from the E_{pc} approach are still comparable across different RX substrates. That said, it does not mean that for a specific RX substrate, k_0 derived from the E_{pc} approach would have a ~10% standard deviation. For clarity, we have revised a sentence in Supplementary Note 10 on page S56 of the SI: “*Although there exists a systematic error between the two approaches, k_0 values determined by the E_{pc} approach are still comparable across different RX substrates.*”

(c) Whether human intervention is needed to assign the linear fitting.

In this work, human intervention is needed to assign the linear fit of E_{pc} versus $\log_{10}[\text{RX}]$, and we assigned a fixed slope of 0.02958 (which is the value for $\frac{RT \ln(10)}{2F}$ at 25 °C, see Eq. R2 and Supplementary Note 10) for the linear fit in the high [RX] regime with a noticeable anodic shift of E_{pc} . We acknowledge that algorithms that can autonomously assign the linear regime and detect outliers for the linear fit need to be developed in the future to achieve full autonomy of self-driving platforms.

10. This work is a remarkable addition to the field, but any automated mechanistic analysis of reaction mechanisms or in general, the use of mechanistic models in electrochemistry can easily be used in cases where the models break or are not suitable. This is particularly true for autonomous workflows and ML models trained on specific subsets of (electro)-chemical reactivity. Reactions falling into the E/EC/CE/DISP etc subspace of electrochemical reactivity can be rare. In my opinion, a paragraph and critical discussion on the shortcomings

of the model, limitations and underlying assumptions is thus indispensable in the main text of the manuscript.

We appreciate the reviewer's comment and we overall agree with the reviewer's sentiment. Per reviewer's request, the following paragraph has been added to the end of the "Results and Discussion" section.

"The performance of our demonstrated autonomous platform relies on the accuracy and general applicability of the DL-enabled automatic mechanism discernment. The DL model deployed here was trained by simulated voltammograms corresponding to only five classical mechanisms and was tasked for mechanism classification with the assumption that there is only one redox event.³⁴ A more advanced DL model capable of both redox-event detection and mechanistic classification out of a larger library of nine mechanisms, dubbed as EchemNet, has recently been reported to accommodate voltammograms with in principle arbitrary number of redox events.⁵⁵ Such advancement removes the restriction of pre-assigning a potential window for cyclic voltammetry measurements and will engender a higher level of autonomy in experimentation platforms. Nonetheless, advanced DL architectures are still needed to either account for mechanisms out of the existing trained ones or if possible accommodate all the possible mechanistic variations. The intrinsic issue of mechanistic ambiguity in voltammetry, that multiple mechanisms may exhibit similar if not the same voltammetric responses,³⁵ suggests inevitable sampling bias during DL model's training. Thus, credible experimental data with proper mechanistic annotations should be welcomed to further refine the trained DL model and it is necessary to incorporate chemistry insights to further ascertain mechanism assignment."

References:

- (34) Hoar, B. B. et al. Electrochemical mechanistic analysis from cyclic voltammograms based on deep learning. *ACS Meas. Sci. Au* **2**, 595–604 (2022).
- (35) Sun, J. & Liu, C. What and how can machine learning help to decipher mechanisms in molecular electrochemistry? *Curr. Opin. Electrochem.* **39**, 101306 (2023).
- (55) Hoar, B. B. et al. Object-detecting deep learning for mechanism discernment in multi-redox cyclic voltammograms. *ChemRxiv*, DOI: 10.26434/chemrxiv-2023-r2v1k (2023).

Reviewers' Comments:

Reviewer #1:

Remarks to the Author:

The authors have addressed my comments.

Reviewer #3:

Remarks to the Author:

I want to congratulate the authors again to the excellent work and manuscript, who's quality is only topped by their detailed answer to the reviewers questions.

The chemistry community will have a lot of fun reading, reading this one!